

# Recent advances in measurement techniques for atmospheric carbon monoxide and nitrous oxide observations

Christoph Zellweger[1], Rainer Steinbrecher[2], Olivier Laurent[3], Haeyoung Lee[4], Sumin Kim[4], Lukas Emmenegger[1], Martin Steinbacher[1], Brigitte Buchmann[5]

[1] Empa, Swiss Federal Laboratories for Materials Science and Technology, Laboratory for Air Pollution / Environmental Technology, CH-8600 Dübendorf, Switzerland.

[2] Karlsruhe Institute of Technology (KIT), Institute of Meteorology and Climate Research (IMK-IFU), Garmisch-Partenkirchen, Germany.

[3] Laboratoire des Sciences du Climat et de l'Environnement (LSCE/IPSL), UMR CEA-CNRS-UVSQ, Gif-sur-Yvette, France

[4] National Institute of Meteorological Sciences (NIMS), Seogwipo-si, Jeju-do, Korea

[5] Empa, Swiss Federal Laboratories for Materials Science and Technology, Department Mobility, Energy and Environment, CH-8600 Dübendorf, Switzerland.

Correspondence to: C. Zellweger (christoph.zellweger@empa.ch)

**Abstract.** Carbon monoxide (CO) and nitrous oxide ($N_2O$) are two key parameters in the observation of the atmosphere, relevant for air quality and climate change, respectively. For CO, various analytical techniques have been in use over the last few decades. In contrast, $N_2O$ was mainly measured using gas chromatography (GC) with electron capture detector (ECD). In recent years, new spectroscopic methods have become available which are suitable for both CO and $N_2O$. These include Infra-Red (IR) spectroscopic techniques such as Cavity Ring Down Spectroscopy (CRDS), Off-Axis Integrated Cavity Output Spectroscopy (OA-ICOS) and Fourier Transform Infrared Spectroscopy (FTIR). Corresponding instruments became recently commercially available and are increasingly used at atmospheric monitoring stations. We analyse results obtained through performance audits conducted within the framework of the Global Atmosphere Watch (GAW) quality management system of the World Meteorology Organisation (WMO). These results reveal that current spectroscopic measurement techniques have clear advantages with respect to data quality objectives compared to more traditional methods for measuring CO and $N_2O$. Further, they allow a smooth continuation of historic CO and $N_2O$ time series. However, special care is required concerning potential water vapour interference on the CO amount fraction reported by Near-IR CRDS instruments. This is reflected in results of parallel measurement campaigns, which clearly indicate that drying of the sample air is leading to an improved accuracy of CO measurements with such Near-IR CRDS instruments.





## 1 Introduction

The Global Atmosphere Watch (GAW) Programme of the World Meteorological Organization (WMO) coordinates a network of atmospheric composition observations comprising 31 global stations, more than 400 regional stations, and around 100 contributing stations operated by contributing networks (GAWSIS, 2018). These stations provide long-term observations of

atmospheric greenhouse gases (GHGs) and reactive gases such as carbon dioxide ($CO_2$), methane ($CH_4$), nitrous oxide ($N_2O$) and carbon monoxide (CO), which are essential for understanding the GHG budget, both regionally and globally. To take full profit of these observations, the uncertainty of these measurements must be minimised which implies consistent data series with traceability to common reference standards. Within the GAW Programme, Central Calibration Laboratories (CCLs) provide reference standards that are linked to internationally accepted calibration scales (Rhoderick et al., 2016; Rhoderick et

al., 2018). In addition, World Calibration Centres (WCCs) evaluate GAW stations through independent assessments by on-site system- and performance audits (Buchmann et al., 2009). Empa operates the World Calibration Centre for Carbon Monoxide (CO), Methane ($CH_4$), Carbon Dioxide ($CO_2$) and Surface Ozone (WCC-Empa) since 1996 as a Swiss contribution to the GAW Programme and has conducted over 90 system- and performance audits over the past 20 years. Furthermore, WCC-Empa collaborates closely with the World Calibration Centre for nitrous oxide (WCC-$N_2O$) to increase the number of

$N_2O$ audits. In order to address scientific needs for interpreting regional or global scale atmospheric observations, the GAW Programme sets ambitious compatibility goals, which are continuously reviewed and, if necessary, revised during biannual meetings of the WMO/GAW community (WMO, 2018). Network compatibility goals are set for amount fraction ranges observed in the unpolluted troposphere, while extended compatibility goals reflect the less stringent requirements for urban and regional studies with larger local fluxes. The network compatibility goals currently stand at ±2 nmol/mol for CO and ±0.1

20  nmol/mol for $N_2O$, whilst the extended goals are set to ±5 nmol/mol for CO and ±0.3 nmol/mol for $N_2O$.

In-situ measurements of tropospheric CO and $N_2O$ have been available since the late 1960s (Weiss, 1981; Khalil and Rasmussen, 1983, 1988). While early measurements were mainly analysis results based on flask samples, quasi-continuous measurements have been available since the early 1980s (Brunke et al., 1990). Although continuous measurements of CO and $N_2O$ began approximately at the same time and were often collocated, challenges with respect to the measurement techniques

for continuous measurements were completely different. Carbon monoxide shows a high temporal and spatial variability, whilst the detection of very small changes is needed for $N_2O$ observations. In the past, atmospheric CO and $N_2O$ measurements at remote locations were almost exclusively made by gas chromatographic techniques (GC). GC with an electron capture detector (GC/ECD) was by far the most abundant measurement technique for $N_2O$, whereas flame ionization detection (GC/FID) in combination with a methaniser and GC with mercuric oxide reduction detector (GC/HgO) was used for CO

(Zellweger et al., 2009).

Recent years brought the rapid development of a variety of alternative CO measurement techniques, and a variety of methods is now in use at atmospheric monitoring sites. Common methods include GC techniques (Gros et al., 1999; Novelli, 1999; van der Laan et al., 2009), non-dispersive infrared absorption (NDIR) (Parrish et al., 1994; Nedelec et al., 2003), vacuum ultra-



Atmospheric Measurement Techniques Discussions - Open Access EGU

violet resonance fluorescence (VURF) (Gerbig et al., 1999), Fourier Transform Infrared (FTIR) absorption (Griffith et al., 2012; Hammer et al., 2013a), Near-IR-Cavity Ring Down Spectroscopy (NIR-CRDS) (Chen et al., 2013; Yver Kwok et al., 2015), and systems using Quantum Cascade Lasers (QCL) in the mid-infrared such as Mid-IR CRDS, cavity enhanced off-axis Integrated Cavity Output Spectroscopy (OA-ICOS) (Baer et al., 2002; Provencal et al., 2005) and quantum cascade

tuneable infrared laser direct absorption spectroscopy (QC-TILDAS) (McManus et al., 2015).

Alternatives to GC-ECD for $N_2O$ are not as abundant, but several methods have been proposed in recent years. These include instruments deploying optical techniques in the mid-IR, e.g. cavity ring-down (CRDS) spectroscopy, FTIR, OA-ICOS, QC-TILDAS and difference frequency generation (DFG)-based systems. Lebegue et al. (2016) published a comprehensive overview of these techniques as well as their performance under controlled conditions.

The recent optical techniques for CO and $N_2O$ have clear advantages concerning sensitivity, repeatability, linearity, time response, and temporal coverage, resulting in new measurement setups and calibration strategies. However, only a few published studies comparing spectroscopic techniques with GC systems exist for CO (Zellweger et al., 2009; Zellweger et al., 2012; Ventrillard et al., 2017) and $N_2O$ (Vardag et al., 2014; Lebegue et al., 2016). Such comparisons of traditional and upcoming techniques are crucial for a smooth continuation of multi-decadal time series when introducing new analytical

techniques.

In this paper, we analyse data collected during CO and $N_2O$ performance audits made by WCC-Empa and WCC-$N_2O$ from 2002 through 2017 from the perspective of the used measurement techniques. We further present ambient air CO comparisons made with a NIR-CRDS travelling instrument during WCC-Empa audits and show limitations of the NIR-CRDS technique with respect to water vapour interference. Assessment of atmospheric measurements through parallel measurements with a

travelling instrument is complementary to performance audits with travelling standards and round robin experiments and is thus an essential, valuable quality control measure (Hammer et al., 2013b; Zellweger et al., 2016).

## 2   Experimental

System and performance audits (hereafter only called audit) by WCCs are part of the quality management framework of the GAW Programme (WMO, 2017a). WCC-Empa is the designated WCC for CO (since 1997), and since 2009 a collaboration

between WCC-Empa and the WCC for $N_2O$ hosted by the Karlsruhe Institute of Technology (KIT), Institute of Meteorology and Climate Research (IMK-IFU), exists since WCC-Empa started including $N_2O$ comparisons during station audits. The concept of station audits has been described elsewhere (Klausen et al., 2003; Buchmann et al., 2009; Zellweger et al., 2016). WCCs use two different approaches to conduct performance audits: (i) comparisons of travelling standards (TS), i.e. high-pressure cylinders with known nominal values of CO and $N_2O$ amount fractions, and (ii) parallel measurements using a

travelling instrument (TI). The TS method is widely applied, while the TI concept is used less frequently and limited by WCC-Empa to CO, $CO_2$ and $CH_4$.





### 2.1 Comparisons using travelling standards

The audit concept using TS supplies gases from high-pressure cylinders, usually dry natural air or synthetic air, on the instruments of the audited station. Calibrations of TS against reference standards before and after the station audit ensure traceability to the CCL. The results are then analysed by linear regression of the values measured by the station vs. the reference

values assigned by the WCC. At WCC-Empa, $N_2O$ and CO amount fractions in the TS are calibrated since 2010 by an Aerodyne Quantum Cascade Laser spectrometer (QC-TILDAS-CS, Aerodyne Research Inc., MA, USA). Before that, an AL5001 Vacuum UV Resonance Fluorescence analyser (VURF) (AL5001, Aerolaser GmbH, Germany) was used for CO calibrations. Both instruments are described in more detail in Zellweger et al. (2012). Amount fractions are assigned to the TS using a set of several reference standards purchased from the CCL, which is run by the National Oceanic and Atmospheric

Administration / Earth System Research Laboratory (NOAA/ESRL). WCC-$N_2O$ uses a set of TS traceable to a set of secondary standards with direct amount fraction assignments by the CCL. Only comparisons involving instruments calibrated to the same calibration scale used by the WCCs are presented hereafter. For $N_2O$, the calibration scales in use were SIO-98 (Prinn et al., 2000) for audits before 2005, and WMO-X2006 and X2006A (Hall et al., 2007; NOAA, 2018c) afterwards. CO refers to the WMO-X2004, X2014 and X2014A (NOAA, 2018a) calibration scales.

We analysed WCC-Empa performance audit results based on the TS method for carbon monoxide (2005-2017) and nitrous oxide (2009-2017), as well as results of $N_2O$ audits conducted by WCC-$N_2O$ (2002-2013). Details on analytical techniques and instruments of these audits are summarised in Table 1 for CO and Table 2 for $N_2O$. CO audits made by WCC-Empa before 2005 were not considered for the comparison due to the following reasons: (i) stations and WCC-Empa were often not referring to the same CO calibration scale. WCC-Empa was using the WMO-X2000 carbon monoxide scale, while many GAW stations

were still reporting on the older WMO-X88 scale (Novelli et al., 2003) or other scales. (ii) WCC-Empa at that time based its calibration of travelling standards only on CO standards above 185 nmol/mol; the WMO-2000 calibration scale had linearity issues, which have been corrected by the succeeding WMO-X2004, X2014 and X2014A calibration scales. For CO, the assessment has been made in the same standardised way as for carbon dioxide ($CO_2$) and methane ($CH_4$) described in Zellweger et al. (2016), while a slightly different approach has been chosen for $N_2O$ due to the fact that ambient air amount fractions

increased significantly during the period of observation. The results section gives further details on the methodology.

### 2.2 Ambient air comparisons

Assessments based on TS comparison, e.g. during station audits or round robin experiments, have limitations. They only cover the analytical system and exclude other aspects that might also be relevant, such as inlet or drying systems. The low water content of the TS may for example lead to a systematic bias, especially for analysers based on spectroscopic techniques with

implemented water vapour correction algorithms. The assessment during on-site audits should therefore include parallel measurements with a TI whenever feasible (WMO, 2011, 2012, 2014, 2016).





WCC-Empa implemented this additional approach for CO, $CO_2$ and $CH_4$ audits. Details of the set-up and procedure as well as results for $CO_2$ and $CH_4$ are published in Zellweger et al. (2016). Audits involving parallel measurements for CO were conducted using a NIR-CRDS analyser (G2401, Picarro Inc., USA) as a travelling instrument. The Picarro G2401 instrument has an internal water vapour correction for CO and reports dry air amount fraction only. However, these factory based

corrections are often not adequate (Chen et al., 2013). Due to the higher analytical noise compared to $CO_2$ and $CH_4$ measurements corrections require a more comprehensive approach (Rella et al., 2013).

The internal water vapour correction of the TI was evaluated using the water droplet method (Zellweger et al., 2012; Rella et al., 2013). Approximately 0.8 ml of ultra-pure water is injected into a constant flow of about 500 ml min$^{-1}$ of a dry working standard and delivered to the instrument using a bypass overflow. For the WCC-Empa CO analyser, the water vapour influence

on the CO amount fraction, which is already corrected by the internal water vapour compensation of the Picarro instrument, was then fitted by a quadratic function. Due to the relatively large uncertainties of individual experiments, we were not able to determine a reliable correction function and, therefore, relied on the factory settings for our experiments.

Parallel measurements with the TI of the following GAW stations are shown in this paper:

    (i)     Puy de Dôme (PUY), France, a global GAW station that is part of the European Integrated Carbon Observation

System (ICOS). A separate inlet system leading to the same location as the air intake of the station analyser was in place for the comparison with the TI. An additional pump at a flow rate of approximately 2 L/min flushed this WCC-Empa inlet line. For the last days of the comparison, the TI sampled from the station inlet using the same cryogenic dryer as the station instrument. During this period, the air was dried to a dew point of approximately -50°C.

(ii)    Anmyeon-do (AMY), South Korea, a regional GAW station run and managed by the Environmental Meteorology Research Division of the National Institute of Meteorological Sciences (NIMS). Air was taken with both instruments from the AMY air inlet system, and the air was dried to a dew point of approximately -50°C using a cryogenic trap.

Table 3 gives an overview of the comparisons, including duration and instruments used. Detailed information about the stations

is available from the GAW Station Information System (GAWSIS, 2018).

## 3    Results

### 3.1    Analysis of travelling standard comparison

One of the objectives of this work was to evaluate the performance of instruments for measuring CO and $N_2O$ at remote atmospheric research observatories. Of particular interest is the question if modern spectroscopic techniques such as NIR-

CRDS, TILDAS, OA-ICOS or FTIR have a significant advantage compared to traditional methods, and whether spectroscopic techniques improve the results of the performance audits carried out by the WCCs for the corresponding compounds with respect to precision and uncertainty. WCC-Empa made sixty comparisons during station audits using travelling standards for



CO (2005-2017), and twenty for $N_2O$ (2009-2017). In addition, WCC-$N_2O$ conducted sixteen comparisons during station audits (2002-2013). Table 1 and 2 show details of analytical techniques and instruments of these comparisons for CO and $N_2O$, respectively. The three letter codes (GAW ID) refers to the different stations (GAWSIS, 2018). Results of audits at the central calibration facility run by the Centre for Atmosphere Watch & Services (CAWAS) and of the greenhouse gas analysis on-

board the Civil Aircraft for the Regular Investigation of the atmosphere Based on an Instrument Container (CARIBIC) are also included in the comparisons.

Each of the audits shown in Table 1 and 2 involved the comparison of a set of travelling standards and was then evaluated by linear regression analysis of the measured values by the stations vs. the WCC assigned amount fractions, which are traceable to the CCL. To judge whether the combinations of the resulting slope and intercept meet the WMO/GAW compatibility,

respectively extended compatibility goals, the method described in Zellweger et al. (2016) was applied in analogy. For CO, the bias at 165 nmol/mol, which is the centre of the amount fraction range of 30-300 nmol/mol representing the unpolluted troposphere (WMO, 2018) was plotted against the slope of the individual travelling standard comparisons. This amount fraction range sufficiently covers the inter-hemispheric gradient, year-to-year variability, seasonal cycles as well as observed trends for the period of consideration at remote stations. For $N_2O$, using a fixed amount fraction range however might not be

appropriate due to the significant upward trend in the atmosphere over the past decades. The range currently representing the unpolluted troposphere has been recently identified as 325-335 nmol/mol (WMO, 2018), which corresponds well to the mean global atmospheric $N_2O$ amount fraction of $328.9 \pm 0.1$ nmol/mol observed in 2016 (WMO, 2017b). A trend analysis made by Blunden and Arndt (2017) showed an annual increase of about 0.8 nmol/mol per year over the last decade, which is in agreement with a fairly constant annual growth rate of 0.81 nmol/mol per year from 1977 until today determined by the

National Oceanic and Atmospheric Administration (NOAA, 2018b). Based on this, our analysis of $N_2O$ audit results was made using a variable amount fraction range covering 10 nmol/mol with the centre being representative for the unpolluted troposphere for the year of the audit. Table 2 gives the corresponding ranges used for the analysis. This method allows displaying the result of each individual CO and $N_2O$ audit involving comparisons with travelling standards as a single dot in a bias vs. slope plot, similar to $CO_2$ and $CH_4$ results presented by Zellweger et al. (2016).

### 3.1.1    Evaluation of CO comparisons

Figure 1 shows the bias in the centre of the relevant amount fraction of the unpolluted troposphere of 30 – 300 nmol/mol CO vs. the slope for the CO audits listed in Table 1. The allowed bias / slope combinations meeting the compatibility (green area) and extended compatibility goals (yellow area) of ±2 nmol/mol and ±5 nmol/mol (WMO, 2018), respectively, are indicated. The distribution of the observed biases and slopes gives further information about potential systematic offsets, which could be

present either at the WCC or at the stations. If results are not systematically biased (e.g. by different calibration scales), a normal distribution of the observed bias and slope pairs around 0 nmol/mol (bias) and 1.0 (slope) is expected. This was the case at the 95% confidence level for the slope, which deviated with a mean value of $0.994\pm0.068$ (1σ) not significantly from one ($t$-Test, $p = 0.47$). However, the mean bias of $-2.6\pm8.7$ nmol/mol (1σ) was significantly different from zero ($p = 0.02$). A



potential reason could be upward drift in standards, which is common for CO in air mixtures at ambient amount fractions (Novelli et al., 2003; Gomez-Pelaez et al., 2013). Drift rates are usually on the order of up to one nmol/mol per year. To account for this, WCC-Empa frequently retrieves reference standards from the CCL. This might not always be the case at measurement sites. There, standards are often in use over long periods without re-calibration or acquisition of new standards.

The use of standards having increased amount fractions due to drift for instrument calibration will then results in an underestimation of ambient CO, which potentially explains the observed mean bias.

Figure 1 shows that reaching the compatibility goals for CO is extremely challenging. The variety of measurement techniques is quite large with clear performance differences between methods. Newer spectroscopic techniques such as QCL based TILDAS or OA-ICOS spectroscopy (QCL hereafter), or CRDS generally show better performance compared to GC methods

or NDIR. Moreover, they also yield higher data coverage due to the truly continuous observations in contrast to the semi-continuous GC measurements and the less frequently required application of reference gases compared to NDIR measurements. Higher data coverage further reduces the uncertainty caused by incomplete sampling. Figure 2 summarises the percentage of comparisons that met the compatibility and extended compatibility goals for (a) all comparisons (see also Figure, right), (b) for GC/HgO and GC/FID systems only, (c) NDIR instruments only, (d) VURF instruments only, and (e) for NIR-

CRDS and QCL instruments. FTIR is not shown separately, since only two comparisons of one instrument were made. Out of the sixty comparisons, only thirteen (21.7%) met the compatibility goal and an additional fourteen (23.3%) met the extended goal in the amount fraction range relevant for the troposphere. Good performance over the entire relevant amount fraction range is required, since atmospheric CO variability is large and pollution episodes, e.g. through long-range transport, are common even at remote locations. Calibration strategies therefore should cover the entire range, which is easier to implement

for techniques with a linear response such as VURF, NIR-CRDS and QCL. The analysis of the performance audit results shows that 90% of the NIR-CRDS and QCL comparisons were meeting the compatibility or extended compatibility goal, while this was the case for less than 40% of the NDIR analysers or GC systems. From the total of ten travelling standard – NIR-CRDS/QCL comparisons, five (50%) were within ±2 nmol/mol, and additional four (40%) within ±5 nmol/mol. The corresponding numbers are significantly smaller for GC based methods (total 18 comparisons) and NDIR (total 23

comparisons), which clearly indicates an advantage of the recent methods compared to more traditional techniques.

However, these results also depend on calibration and potential issues or differences in the calibration scales. For example, an instrument with perfect repeatability and reproducibility but wrong calibration, e.g. by a bias in the calibration standard, can be outside the quality goals only because of calibration issues. In this case, the uncertainty of the linear regression of the travelling standard comparison is expected to be smaller compared to instruments with poorer repeatability and reproducibility.

Therefore, the uncertainty of the linear regression analysis is another measure of the instrument performance. Figure 3 shows a boxplot of the standard uncertainty of the slopes of all CO performance audits grouped by different analytical techniques. The results also confirm the better performance of the QCL and NIR-CRDS instruments compared to GC techniques and NDIR. Interestingly, the performance of NDIR analysers and GC/HgO systems is similar but likely due to different reasons. While the repeatability of GC/HgO systems is generally superior compared to NDIR, appropriate compensation of the non-



linearity remains obviously difficult compared to the normally linear but noisy NDIR analysers, resulting in similar performance of both techniques in the field for the amount fraction range from 30 to 300 nmol/mol.

Comparison with the the recent WMO/IAEA Round Robin Comparison Experiment, as done for $N_2O$ (see below), is not straightforward. Changes in the calibration scale during the round robin experiment jeopardizes the direct comparison of the

audit results with the round robin results.

### 3.1.2  Evaluation of $N_2O$ comparisons

Figure 4 shows the bias in the centre of the relevant amount fraction of the year of the comparison vs. the slope for the $N_2O$ audits shown in Table 2 along with the allowed bias / slope combinations meeting the compatibility (green area) and extended compatibility goals (yellow area) of 0.1 nmol/mol and 0.3 nmol/mol (WMO, 2018), respectively. Only results of comparisons

made on the same calibration scale and fully functional instruments were considered.

The results presented in Figure 4 show that reaching the WMO/GAW compatibility goals remains difficult for $N_2O$. However, calibration ranges at stations can be intentionally limited to the ambient amount fraction typical for their location and time. These ranges are normally significantly smaller than those used in Figure 4 in the case of $N_2O$. Therefore, bias / slope pairs outside the compatibility goals do not necessarily imply that the measurements at a station are biased, but they are indicative

of the performance of the instrument and its calibration over a given amount fraction range. The dashed green and yellow lines in Figure 4 denote the limits for meeting the compatibility and extended compatibility goals at the relevant amount fraction.

As discussed above for CO, the distribution of the observed biases and slopes is an indicator of potential systematic offsets, either at the WCCs or at the stations. No significant deviations at the 95% confidence level were observed for audits carried out by WCC-Empa, with a mean bias of 0.32±1.09 nmol/mol (1σ), $t$-Test p value of 0.11, and a mean slope of 0.965±0.093 (p

= 0.21). WCC-$N_2O$ comparisons showed also no significant bias (-0.12±0.89 nmol/mol, p = 0.35) but the deviation of the slope was significant (0.954±0.067, p = 0.01). This result indicates that at the launch of the audits in 2002 the linearity problem of the ECD was not fully considered in the data evaluation by the audited stations. The GC/ECD technique, which contributes most to the results, is known to be highly non-linear (Lebegue et al., 2016), and consequently, deviations are expected for amount fractions away from the relevant level if the non-linearity of the systems had not been determined accurately enough.

With ongoing data quality assurance activities and the implementation of linearity corrections for the ECD response the slope now is close to one for more recent performance audits.

Fig. 5 presents the result of the above analysis as percentages of comparisons meeting the compatibility and extended compatibility goals. Until now, none of the performance audits conducted by either WCC-$N_2O$ or WCC-Empa achieved the compatibility goal of 0.1 nmol/mol, and only one third of the results were within the extended goals of 0.3 nmol/mol when an

amount fraction range of 10 nmol/mol is considered. This slightly improves if we consider only the bias at the relevant amount fraction. Under these less stringent conditions, we find 19.4 % compliance with the compatibility goal and 36.1% with the extended compatibility goal. This is in line with the above mentioned small variations in $N_2O$ at remote locations and the corresponding limited calibration range of many stations. Lebegue et al. (2016) recognised that measurements of small



variations in the N$_2$O amount fractions using GC/ECD is very challenging, which is in agreement with the TS comparison results from the station audits of this work.

The results obtained during the performance audits by WCC-Empa and WCC-N$_2$O compare well with the recent WMO/IAEA Round Robin Comparison Experiment organised and coordinated by the CCL for N$_2$O hosted by NOAA. The sixth round

robin experiment took place in 2014/15, and involved the comparison of two standards, one containing a lower (average 321.6 nmol/mol) and the other a higher (average 333.7 nmol/mol) N$_2$O amount fraction (NOAA, 2018d). A total of 25 laboratories participated in this exercise. With this data set, we made the same analysis as described above after the exclusion of two laboratories using other calibration scales than WMO-X2006A. The percentage of laboratories fulfilling the WMO compatibility and extended compatibility goal was very similar to the results from the station audits by WCC-Empa and WCC-

N$_2$O, as shown in Figure 6.

Out of the 25 laboratories in the Round Robin Experiment, only two (8%) were entirely within the WMO/GAW compatibility goal of 0.1 nmol/mol for the 10 nmol/mol range. At the relevant amount fraction, the percentage of laboratories that were not meeting the quality goals was very similar for the WCC audits (44%) and the round robin experiment (40%).

The above results, both for TS comparisons during audits and the round robin experiment, are clearly illustrating that it remains

highly challenging to reach the compatibility and extended compatibility goals for N$_2$O. In contrast to advances made for the detection of CH$_4$, CO$_2$ (Zellweger et al., 2016) and CO, measurements of N$_2$O were in most cases still made based on gas chromatography, and only a few recent comparisons involved spectroscopic techniques. The data for N$_2$O clearly indicates advantages of the spectroscopic techniques compared to gas chromatography. The uncertainty of the observed intercepts and slopes of the linear regression gives information of the linearity and repeatability of the system. The uncertainty of the slope

of the linear regression was significantly smaller for QCL and FTIR analysers (median 0.0028, standard deviation 0.0031) compared to GC/ECD systems (median 0.0126, standard deviation 0.0284). Despite the better performance regarding linearity and repeatability of the spectroscopic techniques compared to GC/ECD, no clear advantage of the spectroscopic methods was observed during the performance audits. A potential reason could be the uncertainty of the calibration standards, which is in case of N$_2$O in the same order or even larger than the WMO/GAW compatibility goal. The CCL determined a reproducibility

of N$_2$O calibrations in the ambient range of ~0.22 nmol/mol (95% confidence level) (Hall et al., 2007; NOAA, 2018c), which is larger than the compatibility goal. However, this uncertainty is low compared to uncertainties associated with gravimetric preparation of standards, which highlights the importance of maintaining and propagating calibration scales (Brewer et al., 2018) as implemented in the WMO/GAW programme. Therefore, it is yet too early to quantify this improved performance of spectroscopic techniques for N$_2$O and give a final statement with respect to the compatibility goals.



## 3.2    Ambient air comparisons

The above results, as well as round robin experiments, are travelling standard comparisons and are therefore not covering all aspects of ambient air measurements. Other aspects include bias due to sampling procedures, drying or – related to it – insufficient accounting of spectral interferences, e.g. by water vapour. For example, Chen et al. (2013) demonstrated that

accurate measurements of CO in humid air is possible with the NIR-CRDS technique implemented by Picarro. Correction functions however are different for each individual instrument, and as a result of the work of Chen et al. (2013), these functions are now implemented in Picarro NIR-CRDS CO analysers after 2012.

WCC-Empa started with parallel measurements of ambient air for CO, $CO_2$ and $CH_4$ during station audits in 2011. The results of the greenhouse gas comparisons showed that additional information, e.g. related to air inlet systems, is obtained by these

comparisons (Zellweger et al., 2016). However, these comparisons were in many cases less conclusive for CO. Some parallel measurements showed differences that were not present in the travelling standard comparisons. Sampling issues were unlikely because the ambient air comparison of $CH_4$ and $CO_2$ agreed well. Therefore, other issues like interferences of ambient air constituents may cause an additional bias.

For example, the comparison made at the global GAW station Puy de Dôme (PUY) in 2016 showed significant deviations in

ambient CO measurements, as illustrated in Figure 7, while the TS comparison showed good agreement. During this period, the TI was measuring on average 5.85±0.94 nmol/mol (1σ) lower than the PUY analyser. Despite this bias, both instruments captured the temporal variation well. The WCC-Empa travelling instrument was sampling from the same air intake location but with a completely independent sampling line. In contrast to the PUY instrument, which sampled air dried to a dew point of -50°C, the air sampled by the travelling instrument was not dried. As discussed in the section 2.2, the factory vapour

correction was used. The observed bias correlates with the measured water vapour, as shown in Figure 8, which indicates issues with the internal water vapour compensation of the TI. Water vapour correction functions of this instrument were determined three weeks before and three weeks after the comparison campaign with a droplet test, in analogy to the method described by Rella et al. (2013). Figure 9 shows the ratio of CO(humid, corrected) / CO(dry) against the measured water vapour content of the TI; CO(dry) is the amount fraction measured by the instrument in the absence of water, and CO(humid,

corrected) the water vapour corrected CO amount fraction reported by the Picarro G2401 during the humidification by the droplet test. Since the Picarro G2401 reports CO only as dry air amount fraction, the measured ratio should be equal to one and not depend on water vapour content. However, a significant change in the CO response in relation to water vapour was observed. The TI was underestimating the CO amount fraction in the experiment before the campaign (Figure 9a), and then changed to an overestimation after the campaign (Figure 9b). Possibly, this has been influenced by the upgrade to a new

software version of the TI between the two periods. Unlike for $CO_2$ and $CH_4$, individual water vapour correction functions for CO can currently not be determined with sufficient accuracy to achieve the WMO/GAW network compatibility goal of 2 nmol/mol. Individual experiments using the droplet test have a large uncertainty due to higher instrumental noise for CO compared to $CH_4$ or $CO_2$. Furthermore, CO correction functions seem to be less stable over time, and sudden changes are



possible. Figure 10 shows fitted ratios of CO(humid, corrected) / CO(dry) vs. the measured water vapour content for two different instruments over a period of several years. Both instruments show significant variation over time in the humidity corrected CO reported by the analyser. Consequently, drying of the sample air could improve CO measurements with Picarro G2401 instruments, and likely with Picarro G1302 and G2302 CO/CO$_2$/H$_2$O analysers. This has been confirmed by a period

of dry ambient air measurements of both instruments at PUY. Figure 11 shows the comparison of the two analysers during the audit collocation measurement. In this case, the TI was connected to the same sampling line as the PUY instrument after the cryogenic trap, and both instruments were measuring dry air. The bias of the TI significantly decreased to -1.20±0.57 nmol/mol (1σ). This agrees well with the observed bias during the travelling standard comparison. Figure 12 is summarising the results of the performance audits at PUY with TS, as well as the bias observed during the comparison campaign with humid and dry

measurement of the TI.

Figure 13 shows another example of a CO ambient air comparison made at the regional GAW station Anmyeon-do, South Korea, over a period of one month in 2017. The comparison was made between the AMY cavity enhanced off-axis Integrated Cavity Output Spectroscopy analyser (LGR-30-EP, Los Gatos Research, USA) and the WCC-Empa Picarro G2401 travelling instrument. Both analysers were measuring ambient air dried to a dew point of -50°C using a cryogenic trap. Temporal

variability at this site is significantly larger compared to PUY, and except for a few spikes, it was well captured by both instruments. The bias of the AMY analyser averaged to 0.23±8.81 nmol/mol (1σ) over the entire period of the campaign. However, during the first third of the campaign, the AMY instrument was slightly underestimating the CO amount fraction compared to WCC-Empa, followed by a slight overestimation in the second third. The last third then showed good agreement between the two systems. These differences are likely due to different calibration strategies. The TI was measuring three

standard gases to calibrate and compensate for drift of the instrument every 30 hours. In contrast, manual calibrations were made of the AMY analyser every 14 days with one calibration standard (dried ambient air traceable to the WMO-X2014A scale) applying as a step-wise change fortnightly, and with no further corrections applied in the meantime. These manual calibrations coincide with the observed change in the bias. Consequently, more frequent calibrations or automated measurements of a working standard to compensate for drift would have further improved the agreement. The ambient air

measurements made at AMY were also in agreement with the TS comparison, which is illustrated in Figure 14. The scatter in the bias is significantly larger for ambient air measurements compared to the TS comparison. Firstly, part of this may be explained by the calibration strategy, as discussed above. Secondly, differences in the response time for both instrument types as well as residence time in the inlet might further add to the observed scatter, especially in case of rapid changes in the CO amount fraction, which frequently occurred at AMY.

Both campaigns show that accurate measurements of CO are possible if the sample air is dried. So far, this has not yet implemented at all measurement stations. The above case study at PUY as well as the experiments done involving the droplet tests only investigated the internally implemented water vapour correction of the Picarro G2401, which proofed to be not sufficiently stable to achieve the network compatibility goals of the WMO/GAW Programme. Alternatively, better determination of the remaining water vapour interference is needed. The droplet method might not be suitable due to the



relatively fast drying process, which results in relative high uncertainties due to the analyser's noise. Alternative methods, e.g. as described by Reum et al. (2019) or as implemented by the ICOS Metrology Lab, which is using a Bronkhorst Vapour Delivery Module (VDM) able to humidify a gas stream from a tank, might give better results. In addition to improvements of the droplet method, alternative ways to compensate for the water vapour dependent CO bias need to be explored. Chen et al.

(2013) showed that the main uncertainty of the water vapour correction is due to the fact that the weak CO absorption line is bracketed by adjacent absorptions of $CO_2$ and $H_2O$. Our results indicate that the compensation of the water vapour interference based on the work of Chen et al. (2013), which has been implemented in Picarro analysers newer than 2012, does not correct appropriately all the bias and may change over time. Therefore, frequent determination of the water vapour interference will be needed to ensure long-term stability of the correction function or to characterise its change over time. Consequently, drying

of the sample air should be considered when measuring CO with a Picarro G2401 instrument.

## 4    Conclusions

The different elements of the WMO/GAW quality management framework, including round robin experiments, performance audits with travelling standards and parallel measurements at stations provide complementary information which are essential for reducing the bias and uncertainty of time series measured by atmospheric research stations.

The assessment of performance audit results of CO and $N_2O$ with respect to different measurement techniques showed clear advantages of newer spectroscopic techniques such as NIR-CRDS or QCL spectroscopy in the case of CO. However, parallel measurements made using a Picarro NIR-CRDS analyser identified issues with the implemented water vapour compensation, and further improvement are currently only possible by drying of the sample air.

For $N_2O$, one of the limitations is the uncertainty of calibration standards. This highlights the importance of maintaining

traceability to an internationally accepted calibration scale as implemented by the GAW programme.

By introducing modern spectroscopic measurement techniques such as CRDS or QCL, the number of GAW stations complying with the WMO/GAW compatibility goals for CO and $N_2O$ will increase. However, reaching the compatibility goal of 2 nmol/mol for CO and 0.1 nmol/mol for $N_2O$ will remain challenging. Careful calibration strategies and appropriate water vapour corrections or drying of the sample air are required for both CO and $N_2O$.

**Data availability.** Data from the performance audits made by WCC-Empa are available from the corresponding audit reports (http://www.empa.ch/web/s503/wcc-empa). Data of the WMO/IAEA Round Robin Comparison Experiment are publicly available on the NOAA Earth System Research Laboratory / Global Monitoring Division webpage (https://www.esrl.noaa.gov/gmd/ccgg/wmorr). Other data used in the paper is available upon request to the corresponding

author.

**Competing interests.** The authors declare that they have no conflict of interest.



**Acknowledgements.** MeteoSwiss supported this work through engagement in the Global Atmosphere Watch programme. We further acknowledge the support by the station staff at various GAW stations during the audits. Haeyoung Lee and Sumin Kim would like to acknowledge financial support from the Korea Meteorological Administration Research and Development

5   Program (KMA2018-00522) for the AMY monitoring activity. Mainly Eckhart Scheel, who sadly passed away in 2013, conducted WCC-$N_2O$ audits. Further, the WCC-$N_2O$ acknowledges the funding of the German Environment Agency.



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



*Table 1: CO performance audits using travelling standards from 2005 to 2017*

| Station / Laboratory | GAW ID | Year | Instrument | Method | Intercept (nmol/mol) | Slope (-) | Bias at 165 nmol/mol CO (nmol/mol) |
|---|---|---|---|---|---|---|---|
| Ryori | RYO | 2005 | Horiba GA-360 | NDIR | 7.0 | 0.989 | 5.1 |
| Mt. Kenya | MKN | 2006 | TEI 48C-TL | NDIR | -4.2 | 0.965 | -10.0 |
| Zugspitze-Schneefernerhaus | ZSF | 2006 | AL5001 | VURF | 2.0 | 0.957 | -5.1 |
| Zugspitze-Schneefernerhaus | ZSF | 2006 | AL5002 | VURF | 1.3 | 0.952 | -6.5 |
| Zugspitze-Schneefernerhaus | ZSF | 2006 | TEI 48C-TL | NDIR | -2.0 | 0.988 | -4.0 |
| Hohenpeissenberg | HPB | 2006 | AL5001 | VURF | 0.6 | 0.995 | -0.2 |
| Hohenpeissenberg | HPB | 2006 | TEI 48S | NDIR | -1.3 | 1.000 | -1.3 |
| Jungfraujoch | JFJ | 2006 | Agilent 6890 | GC/FID | 1.9 | 0.987 | -0.2 |
| Jungfraujoch | JFJ | 2006 | Horiba APMA360 | NDIR | 1.2 | 1.012 | 3.2 |
| Cape Point | CPT | 2006 | RGA-3 | GC/HgO | 2.2 | 0.980 | -1.0 |
| Bukit Kototabang | BKT | 2007 | TEI 48C-TL | NDIR | -1.7 | 0.980 | -5.0 |
| Assekrem | ASK | 2007 | Horiba APMA360 | NDIR | 4.0 | 0.995 | 3.2 |
| Pallas | PAL | 2007 | Agilent 6890N | GC/HgO | 0.2 | 0.979 | -3.3 |
| Barrow | BRW | 2008 | RGA-3 | GC/HgO | -1.7 | 1.006 | -0.7 |
| Bukit Kototabang | BKT | 2008 | Horiba APMA360 | NDIR | 0.6 | 0.932 | -10.7 |
| Mt. Kenya | MKN | 2008 | Horiba APMA360 | NDIR | -4.9 | 1.006 | -4.0 |
| Mt. Kenya | MKN | 2008 | TEI 48C-TL | NDIR | -10.0 | 1.032 | -4.6 |
| Ushuaia | USH | 2008 | TEI 48 | NDIR | -1.7 | 0.957 | -8.8 |
| Ushuaia | USH | 2008 | Horiba APMA360 | NDIR | 0.9 | 0.997 | 0.4 |
| Amsterdam Island | AMS | 2008 | RGA-3 | GC/HgO | 10.3 | 0.834 | -17.0 |
| Izaña | IZO | 2009 | RGA-3 | GC/HgO | -5.6 | 1.032 | -0.4 |
| Izaña | IZO | 2009 | TEI 48C-TL | NDIR | -6.3 | 0.922 | -19.2 |
| Santa Cruz | SCO | 2009 | TEI 48C-TL | NDIR | 0.8 | 0.897 | -16.2 |
| Mt. Waliguan | WLG | 2009 | Agilent 6890 | GC/FID | 8.2 | 0.904 | -7.6 |
| CAWAS | NA | 2009 | Agilent 6890 | GC/FID | 6.9 | 0.910 | -7.9 |
| CAWAS | NA | 2009 | Ametek ta500R | GC/HgO | -24.6 | 1.358 | 34.5 |
| Bukit Kototabang | BKT | 2009 | Horiba APMA360 | NDIR | -0.2 | 0.989 | -2.0 |
| Mace Head | MHD | 2009 | RGA-3 | GC/HgO | 1.2 | 1.006 | 2.2 |
| Lauder | LAU | 2010 | FTIR | FTIR | -0.6 | 0.979 | -4.0 |
| Mt. Kenya | MKN | 2010 | Horiba APMA360 | NDIR | -8.1 | 0.978 | -11.8 |



| Mt. Kenya | MKN | 2010 | TEI 48C-TL | NDIR | -6.2 | 1.054 | 2.8 |
|---|---|---|---|---|---|---|---|
| Cape Point | CPT | 2011 | RGA-3 | GC/HgO | 2.2 | 0.953 | -5.6 |
| Zugspitze-Schneefernerhaus | ZSF | 2011 | AL5001 | VURF | 1.7 | 0.977 | -2.1 |
| Zugspitze-Schneefernerhaus | ZSF | 2011 | AL5002 | VURF | 1.4 | 0.987 | -0.7 |
| Hohenpeissenberg | HPB | 2011 | AL5001 | VURF | 0.9 | 0.999 | 0.8 |
| Bukit Kototabang | BKT | 2011 | Horiba APMA360 | NDIR | 4.5 | 0.909 | -10.5 |
| Pallas | PAL | 2012 | PeakPerformer 1 | GC/HgO | 2.8 | 1.042 | 9.8 |
| Pallas | PAL | 2012 | Picarro G2401 | NIR-CRDS | 1.1 | 1.001 | 1.3 |
| Zeppelin Mountain | ZEP | 2012 | Picarro G2401 | NIR-CRDS | -2.5 | 1.015 | 0.0 |
| Zeppelin Mountain | ZEP | 2012 | RGA-3 | GC/HgO | -3.7 | 1.036 | 2.2 |
| Mt. Cimone | CMN | 2012 | Agilent 6890N | GC/FID | -9.7 | 1.048 | -1.8 |
| Mt. Cimone | CMN | 2012 | TEI 48C-TL | NDIR | 4.4 | 0.945 | -4.7 |
| Capo Verde | CVO | 2012 | AL5001 | VURF | 0.0 | 0.991 | -1.4 |
| Capo Verde | CVO | 2012 | LGR-23d | QCL | -3.8 | 1.023 | 0.1 |
| Mace Head | MHD | 2013 | RGA-3 | GC/HgO | -2.9 | 1.101 | 13.8 |
| Izaña | IZO | 2013 | RGA-3 | GC/HgO | -2.3 | 1.010 | -0.7 |
| Bukit Kototabang | BKT | 2014 | Horiba APMA360 | NDIR | -0.4 | 0.873 | -21.4 |
| Jungfraujoch | JFJ | 2015 | Picarro G2401 | NIR-CRDS | -3.9 | 1.008 | -2.6 |
| Jungfraujoch | JFJ | 2015 | LGR-23r | QCL | -1.4 | 1.008 | 0.0 |
| Ushuaia | USH | 2016 | Horiba APMA360 | NDIR | -1.0 | 1.001 | -0.9 |
| Puy de Dôme | PUY | 2016 | Picarro G2401 | NIR-CRDS | -0.1 | 1.001 | 0.0 |
| CAWAS | NA | 2016 | Picarro G2302 | NIR-CRDS | -4.2 | 1.010 | -2.5 |
| Mt. Waliguan | WLG | 2016 | Picarro G2401 | NIR-CRDS | -6.2 | 1.034 | -0.6 |
| Linan | LAN | 2016 | Picarro G2401 | NIR-CRDS | -2.0 | 1.008 | -0.7 |
| Mt. Waliguan | WLG | 2016 | Agilent 6890N | GC/FID | 0.4 | 0.990 | -1.2 |
| Linan | LAN | 2016 | Agilent 7890A | GC/FID | -17.4 | 1.076 | -4.9 |
| Lauder | LAU | 2016 | FTIR | FTIR | 1.6 | 0.955 | -5.7 |
| Anmyeon-do | AMY | 2017 | TEI48i-TLE | NDIR | -42.2 | 1.080 | -29.1 |
| Anmyeon-do | AMY | 2017 | LGR 30-EP | QCL | 1.3 | 0.996 | 0.7 |
| Jeju Gosan | JGS | 2017 | TEI48i-TLE | NDIR | 16.8 | 0.969 | 11.7 |



*Table 2: N$_2$O audits with travelling standards performed by WCC-N$_2$O (2002 - 2013) and WCC-Empa (2009 - 2017)*

| Station / Laboratory | GAW ID | Year | Instrument | Method | Intercept (nmol/mol) | Slope (-) | Ambient bias (nmol/mol) | Range (nmol/mol) | Audit made by |
|---|---|---|---|---|---|---|---|---|---|
| Zugspitze | ZFS | 2002 | HP6890 | GC/ECD | 56.97 | 0.8109 | -3.11 | 312.7-322.7 | WCC-N$_2$O |
| Schauinsland | SSL | 2002 | HP6890 | GC/ECD | 55.18 | 0.8265 | 0.06 | 312.7-322.7 | WCC-N$_2$O |
| Cape Point | CPT | 2003 | Shimadzu GC-8A | GC/ECD | -3.73 | 1.0113 | -0.12 | 313.5-323.5 | WCC-N$_2$O |
| Zugspitze-Gipfel | ZUG | 2005 | HP6890 | GC/ECD | 0.04 | 1.0017 | 0.57 | 315.1-325.1 | WCC-N$_2$O |
| Jungfraujoch | JFJ | 2006 | Agilent 6890 | GC/ECD | 27.26 | 0.9144 | -0.22 | 315.9-325.9 | WCC-N$_2$O |
| Pallas | PAL | 2007 | HP6890 | GC/ECD | -3.20 | 1.0110 | 0.32 | 316.7-326.7 | WCC-N$_2$O |
| Izaña | IZO | 2008 | Varian-CP-3800 | GC/ECD | 2.67 | 0.9915 | -0.08 | 317.5-327.5 | WCC-N$_2$O |
| CARIBIC | NA | 2008 | HP6890 | GC/ECD | 12.49 | 0.9600 | -0.39 | 317.5-327.5 | WCC-N$_2$O |
| Mt. Cimone | CMN | 2008 | Agilent 6890N | GC/ECD | 43.86 | 0.8663 | 0.74 | 317.5-327.5 | WCC-N$_2$O |
| Izaña | IZO | 2009 | Varian-3800 | GC/ECD | 7.81 | 0.9761 | 0.03 | 318.3-328.3 | WCC-Empa |
| Mt. Waliguan | WLG | 2009 | Aglient 6890 | GC/ECD | 50.84 | 0.8569 | 4.60 | 318.3-328.3 | WCC-Empa |
| CAWAS | NA | 2009 | Agilent 6890N | GC/ECD | 4.52 | 0.9863 | 0.33 | 318.3-328.3 | WCC-Empa |
| CAWAS | NA | 2009 | Agilent 6890N | GC/ECD | -4.27 | 1.0141 | 0.27 | 318.3-328.3 | WCC-Empa |
| Mace Head | MHD | 2009 | HP5890 | GC/ECD | -8.58 | 1.0278 | 0.42 | 318.3-328.3 | WCC-Empa |
| Lauder | LAU | 2010 | FTIR | FTIR | -0.72 | 1.0026 | 0.12 | 319.1-329.1 | WCC-Empa |
| Schauinsland | SSL | 2010 | HP6890 | GC/ECD | 9.66 | 0.9710 | 0.28 | 319.1-329.1 | WCC-N$_2$O |
| Cape Point | CPT | 2011 | Agilent 6890N | GC/ECD | -4.52 | 1.0140 | 0.03 | 319.9-329.9 | WCC-N$_2$O |
| Baring Head | BAR | 2011 | Agilent 6890 | GC/ECD | 0.36 | 1.0009 | 0.65 | 319.9-329.9 | WCC-N$_2$O |
| CARIBIC | NA | 2011 | HP6890 | GC/ECD | 6.69 | 0.9784 | -0.32 | 319.9-329.9 | WCC-N$_2$O |
| Mace Head | MHD | 2012 | HP8590 | GC/ECD | 0.81 | 0.9989 | 0.44 | 320.7-330.7 | WCC-N$_2$O |
| Mt. Cimone | CMN | 2012 | Agilent 6890N | GC/ECD | 125.50 | 0.6166 | 0.62 | 320.7-330.7 | WCC-Empa |
| Capo Verde | CVO | 2012 | LGR 23-r | QCL | 14.68 | 0.9545 | -0.13 | 320.7-330.7 | WCC-Empa |
| Mace Head | MHD | 2013 | HP 5800 II | GC/ECD | -0.38 | 1.0004 | -0.25 | 321.5-331.5 | WCC-Empa |
| Izaña | IZO | 2013 | Varian 3800 | GC/ECD | 3.35 | 0.9889 | -0.27 | 321.5-331.5 | WCC-Empa |
| Anmyeon-do | AMY | 2013 | Agilent 7890II | GC/ECD | 6.03 | 0.9802 | -0.44 | 321.5-331.5 | WCC-N2O |
| Jeju Gosan | JGS | 2013 | Agilent 7890II | GC/ECD | 25.49 | 0.9211 | -0.27 | 321.5-331.5 | WCC-N2O |
| Bukit Kototabang | BKT | 2014 | Thermo IRIS 4600 | DFG | -8.86 | 1.0259 | -0.39 | 322.3-332.3 | WCC-Empa |
| Jungfraujoch | JFJ | 2015 | LGR-23r | QCL | 0.87 | 0.9965 | -0.26 | 323.1-333.1 | WCC-Empa |
| Cape Point | CPT | 2015 | Agilent 6890N | GC/ECD | -16.08 | 1.0519 | 0.96 | 323.1-333.1 | WCC-Empa |
| CAWAS | NA | 2016 | AGILENT 7890A | GC/ECD | 11.84 | 0.9645 | 0.16 | 323.9-333.9 | WCC-Empa |





| Mt. Waliguan | WLG | 2016 | AGILENT 6890N | GC/ECD | 11.86 | 0.9638 | -0.06 | 323.9-333.9 | WCC-Empa |
| Linan | LAN | 2016 | AGILENT 7890A | GC/ECD | 7.26 | 0.9781 | 0.05 | 323.9-333.9 | WCC-Empa |
| Lauder | LAU | 2016 | FTIR | FTIR | -0.07 | 1.0036 | 1.12 | 323.9-333.9 | WCC-Empa |
| Anmyeon-do | AMY | 2017 | Los Gatos 30-EP | QCL | 6.43 | 0.9809 | 0.14 | 324.7-334.7 | WCC-Empa |
| Anmyeon-do | AMY | 2017 | Agilent 7890II | GC/ECD | 30.37 | 0.9059 | -0.65 | 324.7-334.7 | WCC-Empa |
| Jeju Gosan | JGS | 2017 | Agilent 7890II | GC/ECD | -2.95 | 1.0083 | -0.22 | 324.7-334.7 | WCC-Empa |





*Table 3: Overview of ambient air CO comparison campaigns.*

| Location | Coordinates | Start | End | Station instrument | Travelling Instrument |
|----------|-------------|-------|-----|--------------------|-----------------------|
| PUY | 45.7723 N 2.9658 E | 2016-04-11 | 2016-06-22 | Picarro G2401 #CFKADS-2161 dry measurements | Picarro G2401 #CFKADS2098 humid and dry meas. |
| AMY | 36.5383 N 126.3300 E | 2017-07-31 | 2017-09-05 | LGR $N_2O$/CO-30-EP #15-0213 dry measurements | Picarro G2401 #CFKADS2098 dry measurements. |

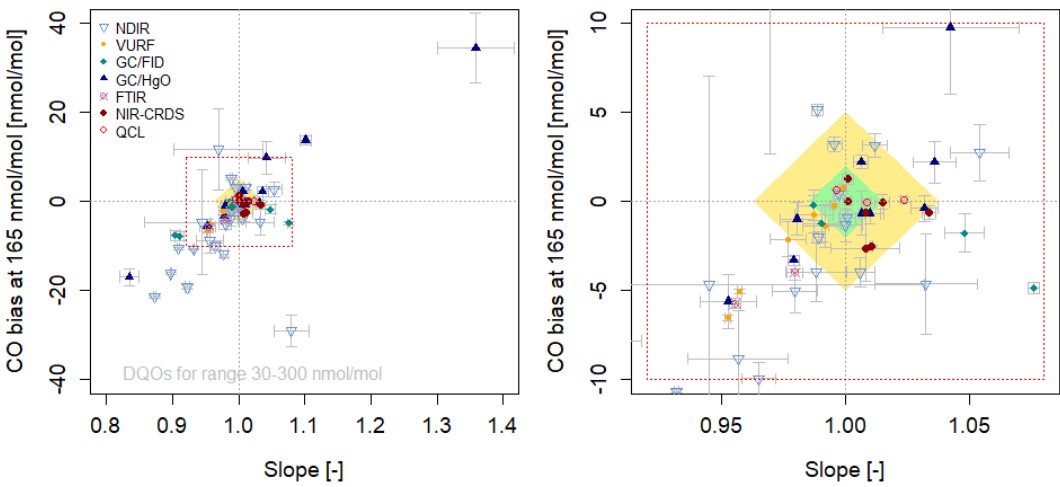

*Figure 1. Left: CO bias at 165 nmol/mol vs. the slope of the audit for individual travelling standard comparisons. Different*
5 *symbols and colours indicate different measurement techniques of the station analysers. The error bars correspond to the*
*uncertainty of the slope and the bias (1σ). The green and yellow areas correspond to the WMO/GAW compatibility and*
*extended compatibility goals for the amount fraction range of 30 - 300 nmol/mol. Right: detail of the red dotted box of the left*
*panel.*





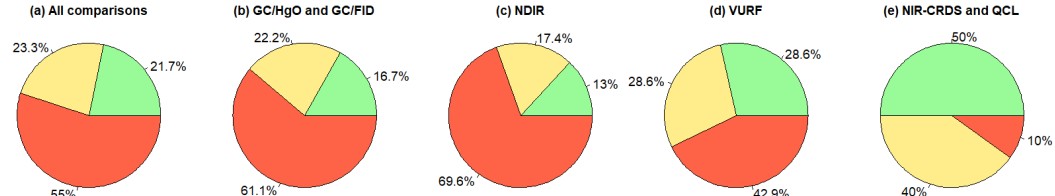

*Figure 2.Percentage of CO performance audit results that were in the range of 30 - 300 nmol/mol within the WMO/GAW compatibility goals (green), the extended compatibility goals (yellow), or outside the compatibility goals (red area) for (a) all comparisons, (b) GC systems, (c) NDIR analysers, (d) VURF analysers, and (e) NIR-CRDS and QCL systems.*

*Figure 3. Boxplot of the slopes uncertainties of the of the regression analysis for the CO performance audits for different analytical techniques. The horizontal blue line denotes to the median, and the blue boxes show the inter-quartile range.*





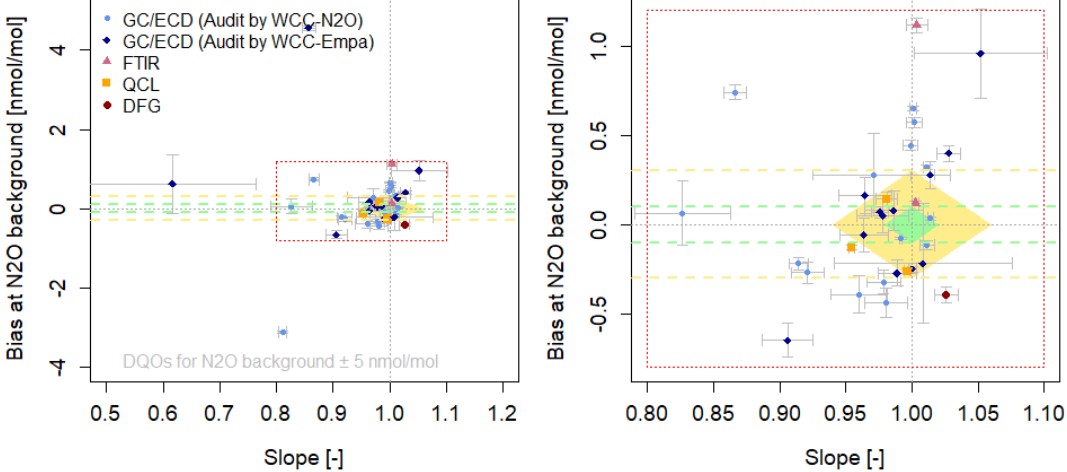

Figure 4. Left: N₂O bias in the centre of the relevant ambient air amount fraction (Table 2) vs. the slope of the audit for individual travelling standard comparisons. Different symbols and colours indicate different measurement techniques of the station analysers. The error bars correspond to the uncertainty of the slope and the bias (1σ). The green and yellow areas

5    correspond to the WMO/GAW compatibility and extended compatibility goals for the range of ± 5 nmol/mol around the centre of the relevant amount fraction range, and the dashed green and yellow lines show the limits at the relevant amount fraction. Right: detail of the red dotted box of the left panel.

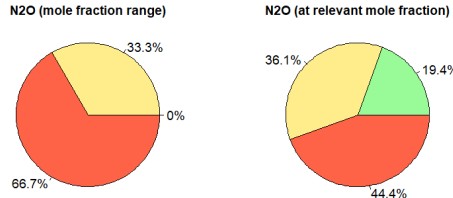

10   Figure 5. Left: Percentage of N₂O performance audit results that were for the range of the relevant amount fraction ±5 nmol/mol within the WMO/GAW compatibility goals (green), the extended compatibility goals (yellow), or outside the compatibility goals (red area). Right: Same as on the left side but at the relevant amount fraction (see text for details).





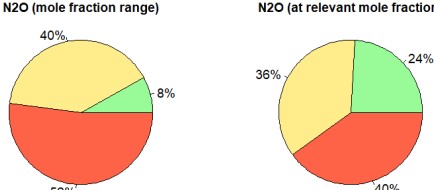

*Figure 6. Left: Percentage of the results of the 6th round robin experiment that were for the range of the relevant amount fraction ±5 nmol/mol within the WMO/GAW compatibility goals (green), the extended compatibility goals (yellow), or outside the compatibility goals (red area). Right: Same as on the left side but at the relevant amount fraction (see text for details).*

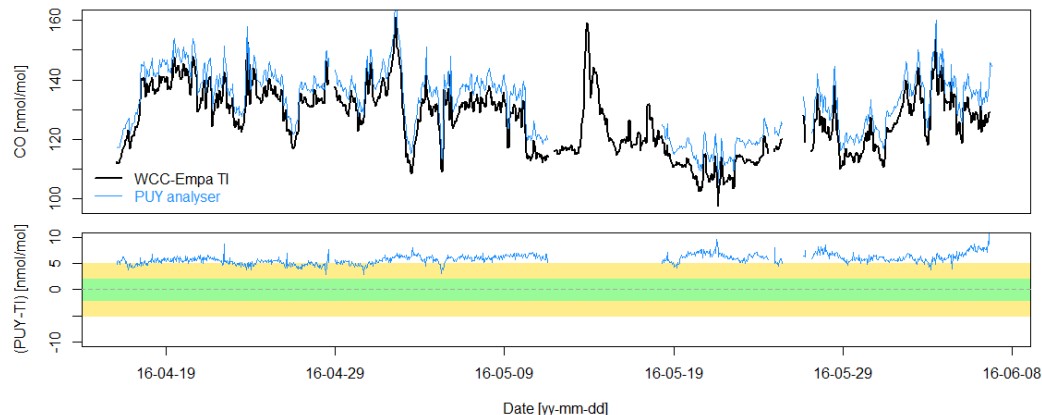

*Figure 7. CO comparison at PUY between the WCC-Empa travelling instrument and the PUY Picarro G2401 for the period when the TI sampled humid air. Upper panel: CO time series (1 h data). Lower panel: CO bias of the station analyser vs time.*
10  *The green and yellow areas correspond to the WMO compatibility and extended compatibility goals.*





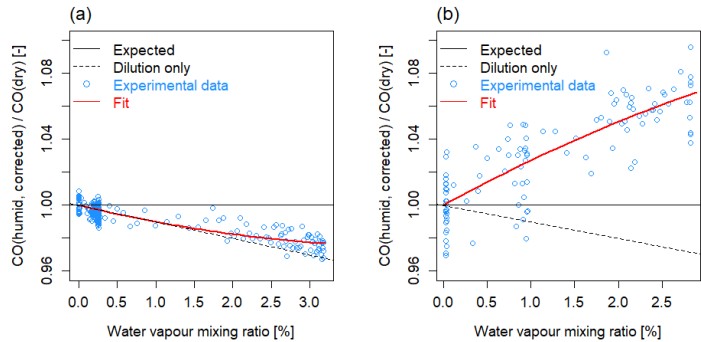

Figure 8. Bias of the PUY Picarro G2401 vs. the water vapour measured by the TI. The solid black line shows the linear regression with 95% confidence interval (dashed lines). The green and yellow areas correspond to the WMO compatibility and extended compatibility goals.

Figure 9. CO(humid, corrected) / CO(dry) vs. the reported water vapour for the experiment before (a) (2016-03-23) and after (b) (2016-07-14) the comparison at PUY.





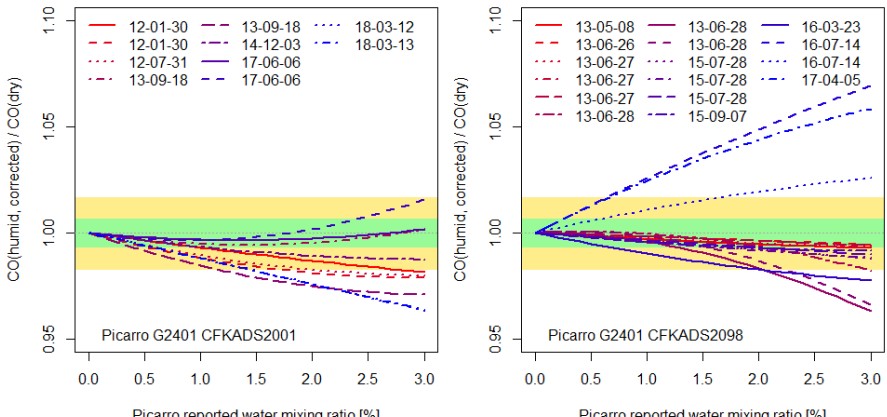

Figure 10. Ratios of CO(humid, corrected) / CO(dry) amount fractions vs. the water vapour mixing ratios of two different Picarro G2401 NIR-CRDS analysers over time. The legend shows the date (yy-mm-dd) of the experiment. The coloured areas are the limits for the WMO/GAW compatibility goal (green) and extended (yellow) compatibility goal at the amount fraction

5 of 300 nmol/mol CO.

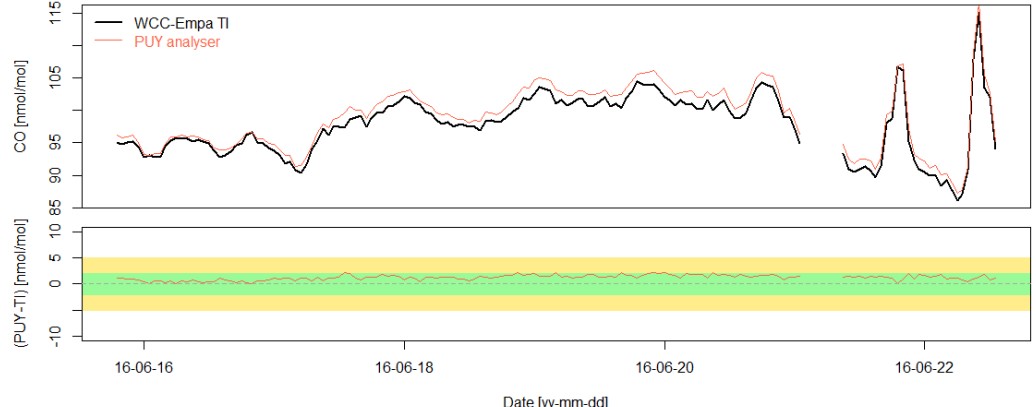

Figure 11. CO comparison at PUY between the WCC-Empa travelling instrument and the PUY Picarro G2401 for the period when the TI sampled dry air. Upper panel: CO time series (1 h data). Lower panel: CO bias of the station analyser vs time.

10 The green and yellow areas correspond to the WMO compatibility and extended compatibility goals.




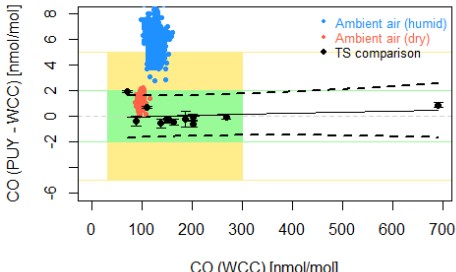

*Figure 12. Bias of the PUY Picarro G2401 CO instrument vs. WCC-Empa assigned values. Black dots represent the average of data at a given level from a specific TS comparison. The error bars show the standard deviation of individual measurement points. The green and yellow lines correspond to the WMO compatibility and extended compatibility goals, and the green and*
5 *yellow areas to the amount fraction range relevant for PUY. The dashed lines around the regression lines are the Working-Hotelling 95percentage confidence intervals. The coloured dots show the bias during the ambient air comparison without (blue) and with (red) drying of the air sampled by the TI.*

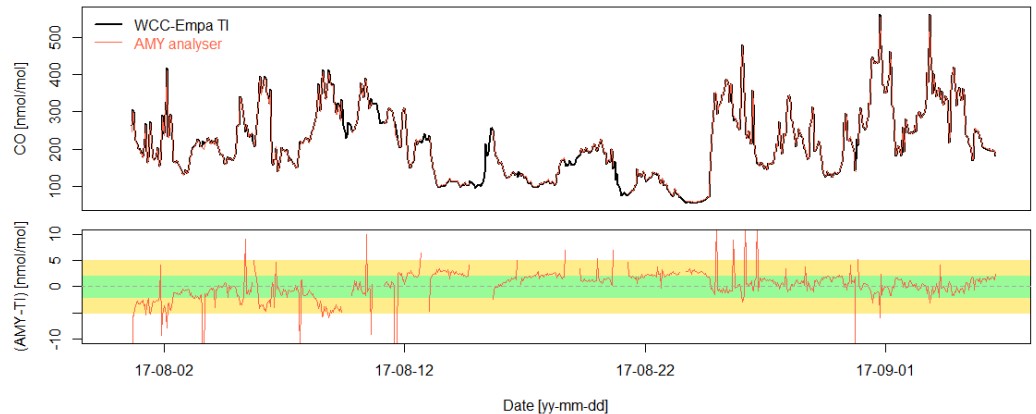

10 *Figure 13. CO comparison at AMY between the WCC-Empa travelling instrument and the AMY Los Gatos 30-EP QCL analyser. Both instruments sampled dry ambient air. Upper panel: CO time series (1 h data). Lower panel: CO bias of the station analyser vs time. The green and yellow areas correspond to the WMO compatibility and extended compatibility goals.*





*Figure 14. Bias of the AMY Los Gatos 30-EP CO instrument vs. WCC-Empa assigned values. Black dots represent the average of data at a given level from a specific TS comparison. The error bars show the standard deviation of individual measurement points. The green and yellow lines correspond to the WMO compatibility and extended compatibility goals, and the green and*

5    *yellow areas to the amount fraction range relevant for PUY. The dashed lines around the regression lines are the Working-Hotelling 95percentage confidence intervals. The red dots show the bias during the ambient air comparison.*