# Peer review of "Recent advances in measurement techniques for atmospheric carbon monoxide and nitrous oxide observations"

_Atmospheric Measurement Techniques, 2019_

## Referee Comment (RC1) · Anonymous Referee #1 · 6 Jun 2019

General comments

The focus of this paper is topical, since the interest and need for better understanding on especially trends of atmospheric concentrations of N2O is raised due the climate change issues during last years. However, the instrumentation and calibration gases are still not at the level enabling the expected accuracy for the measurements of CO and N2O amount fractions. This paper also promotes atmospheric stations to pay attention to the problematics found from the amount fraction measurements by discussing them broadly, and this way improve the quality of atmospheric CO and N2O concentration data globally.

[Figure]

The introduction section describes the background of the field and the aim of the paper shortly but still adequately. Results are dealt with high expertise and there is no doubt that authors have a deep understanding of the field. Since some graphical results are complicated, slightly longer explanations would help the reader, without a need to find a clarification from references. Tables and graphs itself are clear. Authors are referred to the key articles of this topic. The English of this paper is mainly sufficient. Altogether, my opinion is that the scientific quality of this manuscript is high, and it should be published with revisions.

Specific comments

-page 4, section 2.1: there is probably a SOP for the TS measurements at the stations? Like, how the TSs are prepared for the measurement? Is there used a similar sequence for TSs as the station uses for its own cylinder measurements or instructed by WCC? Is the protocol dependent on the instrumentation? Who is measuring the TSs in practice? I am not asking to answer all of these, but just shortly clarifying how the TS measurements done; with a certain fixed protocol or not; since this may affect to the results as well. Stations personnel may have bad habits, but they may also know by experience if their instrument needs special way to handle it.

-page 5, lines 11-12: two questions concerning water vapour correction of WCC. 1) Since droplet method is not giving appropriate results, but in page 12 you are listing two - most probably superior - methods to determine water vapor correction function, why WCC is not using or even testing those methods? 2) If the correction of the effect of water vapour to CO measurements is difficult, why WCC is not drying the sample?

-page 8, lines 27-31/Figure 5: concept "relevant amount fraction" for N2O remains a bit unclear. So, is the "relevant amount fraction range" a central 10 nmol/mol range representative for the unpolluted troposphere for the year of the audit and the "relevant amount fraction" a single representative value $\pm 0.1$ nmol/mol? Please try to state this more explicitly in the text.

[Figure]

-pages 8-9, section 3.1.2: it is clear how and why the compatibility goal for N2O is determined. However, you clearly show in this manuscript that it will not happen in near future that atmospheric N2O measurements will reach the goal. So, should the goal be revisited? I would like to see a statement in conclusions section regarding the necessity of these goals no-one is reaching.

-page 11, lines 1-2/Figure 10: if I understood correctly, the droplet tests are always made with the same CO amount fraction (300 nmol/mol), but is the water vapour response dependent on amount fraction of CO, i.e. have you run the test with other CO amount fractions? It is a bit hard to see from the graph 10; there is no chronological trend, but the water vapour response is varying randomly?

-page 11, lines 4-9: you state that the clearly decreased bias between TI and PUY instrument was thanks to the drying of sample air for TI. However, you also changed the inlet line and never tested the other line with dryer. Please, mention this uncertainty more clearly in the text.

-page 11, line 13: did you use the same individual instrument (TI) in PYU and in AMY? Differences between Picarro G2401 individuals, especially regarding CO performance, may be high. -page 11, lines 16-19: since there was three clearly different periods; due the calibrations of AMY instrument; during the comparison measurements at AMY, please provide the biases for each period separately.

-page 12, line 18: is the drying of the sample air the only option, as you state, or is it possible to reach the goal with well and frequently made water vapour tests as well? You also leave open the questions how to dry and how low water levels are needed to reach. By looking the Figure 10, it looks that water levels below 0.3 %, roughly, would give CO values stable enough when Picarro G2401 instrument is used. That kind of a water levels are possible to reach by using Nafion dryer, for example. The drying issue is an endless story, but open this theme with a few sentences.

-figure 7: since the water vapour was probably the main reason for bad agreement

between TI and PUY instrument, please add a panel showing the H2O % reading of TI.

-figure 13: please, mark the calibrations of AMY instrument in lower panel with vertical lines, for example.

Technical corrections

-page 1, line 20: "analyse" > "analysed".

-page 2, line 1: "GAW programme" is written with and without a capital "p" in this manuscript. Please uniform using lower case "p".

-page 2, line 11: What is "Empa"? Please add institute full name and location.

-page 2, line 11: Use "WCC" instead of "World Calibration Centre". There is often used the full name instead of abbreviation, even though the abbreviation is introduced when first time mentioned. Like three lines below "World Calibration Centre" in written again, and with gas components and instrument techniques is the same issue. Uniform all these issues in the whole manuscript.

-page 2, line 14: WCC-N2O, who operates? It is written on page 3, so move it to here.

-page 2, lines 28-30: introduction of the GC detectors for CO; now it is not fully clear that CG/FID with methaniser and CG/HgO are two separate instruments to measure CO. Please reword.

-page 4, lines 9-10: move the operator of CCL to where it is first time mentioned.

-page 5, line 6: add comma after "measurements".

-page 6, line 15: maybe "trend in the atmospheric concentrations" over "in the atmosphere".

-page 7, lines 13-14: what is the meaning of "(see also Figure, right)"?

---

## Referee Comment (RC2) · Anonymous Referee #2 · 7 Jun 2019

The manuscript "Recent advances in measurement techniques for atmospheric carbon monoxide and nitrous oxide observations" by Zellweger et al. presents comparison data collected through the WMO GAW WCC station audits for CO and N2O. The WMO GAW program has set strict network compatibility goals to ensure data from various monitoring sites and programs are consistent and that biases in the data will not unduly influence scientific interpretation of the data. Network compatibility is a difficult parameter to asses. One tool for assessing it is the station audits conducted by the WCC's. While these are snapshots in time, they are a very rigorous comparison and provide valuable information on the consistency of data from various sites and networks. The data presented in this manuscript is valuable information for

determining the significance of spatial gradients observed when data from various sites and monitoring programs are combined. In addition, the authors have used the comparison data from the audits to clearly show the advantages of newer analytical techniques and to point out areas that need further improvements. This should be of high interest to many involved in atmospheric monitoring. I recommend publication of this manuscript with a few minor suggestions to improve the document. Please see the attached pdf file for specific comments.

Please also note the supplement to this comment:
https://www.atmos-meas-tech-discuss.net/amt-2019-108/amt-2019-108-RC2-supplement.pdf

**Supplement:**

General comments:

The manuscript "Recent advances in measurement techniques for atmospheric carbon monoxide and nitrous oxide observations" by Zellweger et al. presents comparison data collected through the WMO GAW WCC station audits for CO and $N_2O$. The WMO GAW program has set strict network compatibility goals to ensure data from various monitoring sites and programs are consistent and that biases in the data will not unduly influence scientific interpretation of the data. Network compatibility is a difficult parameter to asses. One tool for assessing it is the station audits conducted by the WCC's. While these are snapshots in time, they are a very rigorous comparison and provide valuable information on the consistency of data from various sites and networks. The data presented in this manuscript is valuable information for determining the significance of spatial gradients observed when data from various sites and monitoring programs are combined. In addition, the authors have used the comparison data from the audits to clearly show the advantages of newer analytical techniques and to point out areas that need further improvements. This should be of high interest to many involved in atmospheric monitoring. I recommend publication of this manuscript with a few minor suggestions to improve the document.

Specific comments:

1) Throughout the document – the WMO network compatibility goals should be referred to as "network compatibility" rather than just "compatibility" to distinguish the WMO usage from a strict metrological definition of compatibility. Or making the distinction upon first usage if that is preferred.

2) Page 2, line 11: Define Empa on first usage.

3) Page 2, line 14: Define who is the WCC-$N_2O$ on first usage.

4) Page 4, line 10: "WCC-$N_2O$ uses a set of TS traceable to a set of secondary standards …" Are these standards truly secondary (compared directly to the primary standards) or are they actually the normal tertiary standards distributed by the CCL? Or does this "secondary" label relate to the hierarchy internal to the WCC-$N_2O$?

5) Page 4, line 11-14: Have all of the comparisons been reprocessed onto the current CO_X2014A and $N_2O$_X2006A scales or are they presented on the current scale at the time of the comparison? If the latter is the case I would suggest making the scale explicit by adding a column to tables 1 and 2 showing which scale was actually used. As a follow up I would ask if the comparisons change significantly if reprocessed onto the current scale? This may not be possible within the scope of this paper but would be of interest to data users who would like to use this information to understand potential biases between data from various providers.

6) Page 5, line 1: When did the parallel measurement approach begin?

6) Page 6, line 28: WMO network compatibility goals are no longer listed as "±".

7) Page 7, line 3-6: Have any of the comparisons been reprocessed after the stations have had working standards re-calibrated and drift corrected? It would be very interesting to see if some of these larger offsets are improved with better calibrated standards. This would also provide a more accurate assessment of the bias in the station data but again may be beyond the scope of this paper.

8) Page 8, line 21: Is this statement supported by meta data from the stations, i.e. is there a record of the number of standards used for those early audits that would support this conclusion?

9) Page 11, line 7: The figure plots the data as Station – TI, the text has the offset as TI – Station. I suggest changing either the sense of the comparison in this sentence or in the figure to be consistent.

10) Page 11, line 12: I think the description of the AMY offaxis_ICOS instrument should be "enhanced performance" off-axis integrated cavity output spectroscopy rather than the stated "cavity enhanced".

11) Page 11, line 17-19: I suggest putting in the values for each third of the time period to show how different they are and how much of the variability is due to the calibrations.

12) Page 17, Table 1: I suggest listing the CO scale for each comparison if they are not all the same.

13) Page 19, Table 2: I suggest listing the $N_2O$ scale for each comparison if they are not all the same.

14) Page 22, Figure 3: As mentioned in the text there are only 2 comparisons of the same FTIR instrument. It might be good to show this by listing n values for each category or at least for the FTIR. I might also suggest keeping the same categories at shown in figure 2 (combining NIR-CRDS and QCL) to be consistent between the two figures but leave this to the author's discretion.

15) Page 24, Figure 7 caption (and other time series plots): The caption says "(1 h data)". I take this to mean the data from both instruments was averaged to hourly averages. If true I suggest making this point clearer.

Minor Technical corrections:

The following are places where I feel the writing could be improved by slight wording changes, however this is subjective and I leave it to the author's discretion.

) Page 1, line 26: … clearly indicate that drying of the sample air  "leads" to an improved …

) Page 2, line 6:  "To make full use" of these observations, …

) Page 2, line 25-26: Carbon monoxide shows  high temporal and spatial variability, whilst the detection of very small changes is needed for $N_2O$ observations "due to its low variability."

) Page 2, line 31-32: … and a variety of methods  "are" now in use at atmospheric monitoring sites.

) Page 3, line 10:  The  "recently developed" optical techniques for …

) Page 3, line 13-14:  Such comparisons of traditional and  "new" techniques are crucial for …

) Page 3, line 24-26:  Empa is the designated WCC for CO (since1997), and since 2009 a collaboration between WCC-Empa and the WCC for $N_2O$ hosted by the Karlsruhe Institute of Technology (KIT), Institute of Meteorology and Climate Research (IMK-IFU),  "has allowed" WCC-Empa  "to include" $N_2O$ comparisons during station audits.

) Page 4, line 31: Include the most recent version of the GGMT recommendations "WMO, 2018" in the list of references.

) Page 6, line 9-10: To judge whether the combinations of the resulting slope and intercept meet the WMO/GAW compatibility "or" extended compatibility goals, the method …

) Page 6, line 32:  … which deviated with a mean value of 0.994 ±0.068 (1σ) not significantly "different" from one …

) Page 8, line 20:  WCC-$N_2O$ comparisons "also" showed  no significant bias …

) Page 9, line 8:  … laboratories using  calibration scales "other" than WMO-X2006A.

) Page 9, line 19:  … slopes of the linear regression gives information  "on" the linearity and …

) Page 9, line 18-19: … which is in "the" case of $N_2O$  "of" the same order or even larger than …

) Page 10, line 19:  As discussed in the section 2.2, the factory "water" vapour correction …

) Page 11, line 30:  So far, this has not yet "been" implemented at all …

) Page 11, line 32:  … which  "proved" to be …

) Page 12, line 1:  … which results in  "relatively" high uncertainties due to …

) Page 12, line 2-3:   … which  "uses" a Bronkhorst Vapour Delivery Module (VDM)  to humidify a gas stream from a tank, might …

) Page 12, line 6:  … bracketed by absorption "lines" of $CO_2$ and $H_2O$.

---

## Author Comment (AC1) · 17 Sep 2019

**Referee #1**

We would like to thank Referee #1 for the valuable comments and her/his time to review the manuscript. The questions/comments/concerns of Referee #1 (bold text) are addressed below. New text in the manuscript is given in italics.

**General comment made by Referee #1:**

**The introduction section describes the background of the field and the aim of the paper shortly but still adequately. Results are dealt with high expertise and there is no doubt that authors have a deep understanding of the field. Since some graphical results are complicated, slightly longer explanations would help the reader, without a need to find a clarification from references. Tables and graphs itself are clear. Authors are referred to the key articles of this topic. The English of this paper is mainly sufficient. Altogether, my opinion is that the scientific quality of this manuscript is high, and it should be published with revisions.**

Thank you for this comment. We agree that especially the bias / slope plots might be difficult to understand without further explanation. We refer to Zellweger et al. (2016) for a detailed description of the methodology, including an illustrative plot showing different bias / slope combinations. We believe that duplication of this explanation is not needed here. However, we added a sentence to clarify on page 6 / line 27: "*Perfect agreement would result in bias / slope pairs of (0 nmol/mol / 1)*".

**Specific comments made by Referee #1:**

**-page 4, section 2.1: there is probably a SOP for the TS measurements at the stations? Like, how the TSs are prepared for the measurement? Is there used a similar sequence for TSs as the station uses for its own cylinder measurements or instructed by WCC? Is the protocol dependent on the instrumentation? Who is measuring the TSs in practice? I am not asking to answer all of these, but just shortly clarifying how the TS measurements done; with a certain fixed protocol or not; since this may affect to the results as well. Stations personnel may have bad habits, but they may also know by experience if their instrument needs special way to handle it.**

Yes, there is a SOP for TS measurements at stations, which is available from the WCC-Empa webpage (https://www.empa.ch/documents/56101/250799/2.pdf/f5a8c0a5-884f-4e0c-b836-96d92dbd260c). However, there is no fixed protocol, since this depends on both the instrumentation and the capabilities of the stations. Usually, station operators are analysing the TS as unknown samples on their system, with at least three repetitions for each TS. It is clear that this approach has limitations, and additional parallel measurements with a fully independent analytical system are desirable and are implemented whenever possible during station audits. Brewer et al. (2019) recently showed that results of the audit approach by WCC-Empa are comparable to the WMO round robin experiments (NOAA, 2018). This confirms that the protocol followed by WCC-Empa is a valid approach to assess instrument performance. The limitations of the TS method are listed in section 2.2. We added the following sentence in section 2.1:

*Usually, multiple analysis of a set of three or more TS is made and averaged for the final assignment of the TS value by the audited laboratory.*

**-page 5, lines 11-12: two questions concerning water vapour correction of WCC. 1) Since droplet method is not giving appropriate results, but in page 12 you are listing two - most probably superior - methods to determine water vapor correction function, why WCC is not using or even testing those**

**methods? 2) If the correction of the effect of water vapour to CO measurements is difficult, why WCC is not drying the sample?**

1) It is true that the droplet method has limitations regarding the determination of the water vapour interference for CO. The methods described later in the paper were not yet published / available when the experiments done for this paper were made. Potentially, they would give better results, which would allow for a better compensation of the bias. However, a large part of the uncertainty is due to relatively large instrumental noise of the Near-IR-Cavity Ring Down Spectroscopy (NIR-CRDS) technique, which makes the determination of the appropriate correction function challenging. Furthermore, unpredictable short-term changes as shown in section 3.2 / Figure 10 further complicate appropriate correction.

2) As a consequence of the results, we are now drying the air, which is also one of the recommendations of this paper (see conclusions).

**-page 8, lines 27-31/Figure 5: concept "relevant amount fraction" for N$_2$O remains a bit unclear. So, is the "relevant amount fraction range" a central 10 nmol/mol range representative for the unpolluted troposphere for the year of the audit and the "relevant amount fraction" a single representative value ±0.1 nmol/mol? Please try to state this more explicitly in the text.**

Yes, this is correct. The amount fraction range is given in Table 2 for all audits. We agree that the description of the concept regarding the representative amount fraction was not clear in the manuscript. The relevant amount fraction is a single value and depends on the year of the audit. It was calculated using the value of the unpolluted air in 2016 and an annual growth rate of 0.8 nmol/mol. This value corresponds to the centre of the amount fraction range given in Table 2.

We made the following changes in the manuscript to better explain the concept:

Page 8, line 7: added "*range*".

Page 8, line 31: we added "*The relevant amount fraction corresponds to the value at the centre of the relevant range for the corresponding year.*"

**-pages 8-9, section 3.1.2: it is clear how and why the compatibility goal for N$_2$O is determined. However, you clearly show in this manuscript that it will not happen in near future that atmospheric N$_2$O measurements will reach the goal. So, should the goal be revisited? I would like to see a statement in conclusions section regarding the necessity of these goals no-one is reaching.**

We added the following sentences (from the report of the 19[th] WMO/IAEA Meeting on Carbon Dioxide, Other Greenhouse Gases and Related Tracers Measurement Techniques (GGMT-2017)) after the definition of the network compatibility goals in the introduction (page 2, line 20ff): "*These goals represent the maximum bias that can generally be tolerated in measurements of well-mixed background air used in global models to infer regional fluxes. Some network compatibility goals may not be currently achievable within current measurement and/or scale transfer uncertainties. However, they are targeted for applications which require the smallest possible bias among different datasets or data providers, such as for the detection of small trends and gradients (WMO, 2018).*"

We believe that it is beyond the scope of this manuscript to make recommendations regarding the compatibility goals, since this is done in the larger scientific community during the GGMT meetings.

**-page 11, lines 1-2/Figure 10: if I understood correctly, the droplet tests are always made with the same CO amount fraction (300 nmol/mol), but is the water vapour response dependent on amount**

**fraction of CO, i.e. have you run the test with other CO amount fractions? It is a bit hard to see from the graph 10; there is no chronological trend, but the water vapour response is varying randomly?**

We would like to thank the reviewer for this comment, since it made clear that our description of the experiment was not detailed enough. The water vapour experiments were made at different amount fractions ranging from 57 to 741 nmol/mol CO, but the limits shown in Figure 10 refer to an amount fraction of 300 nmol/mol. We found no dependency of the water vapour correction function on the amount fraction for both instruments, which is shown in Figure R1 where the ratios of CO(humid, corrected) / CO(dry) at a water level of 3% are plotted against the CO amount fractions used for the corresponding determination of the correction function. The limits for the WMO/GAW compatibility goal (green) and extended (yellow) compatibility goals are also shown.

[Figure]

*Figure R1. Black dots show the ratios of CO(humid, corrected) / CO(dry) amount fractions at a water level of 3% vs. the CO amount fraction of the CO standards used during the corresponding experiments of two different Picarro G2401 NIR-CRDS analysers. The coloured lines show the limits for the WMO/GAW compatibility goal (green) and extended (yellow) compatibility goal.*

We also were not able to see a clear temporal trend in the change of the correction function, which is illustrated in Figure R2 below where the ratios of CO(humid, corrected) / CO(dry) at a water level of 3% are plotted against the time in weeks since the first determination of the correction function. The limits for the WMO/GAW compatibility goal (green) and extended (yellow) compatibility goals are also shown in Figure R2 as horizontal lines for an amount fraction of 300 nmol/mol CO. It looks like the Picarro G2401 CFKADS2098 gets less stable after approximately 150 weeks. However, both instruments were not able to fulfil the WMO/GAW network compatibility right from the beginning at a water level of 3%. This clearly supports the conclusion of our work that the internal water vapour compensation is inappropriate, and drying of the air sample is recommended.

[Figure]

*Figure R2. Black dots show the ratios of CO(humid, corrected) / CO(dry) amount fractions at a water level of 3% vs. time in weeks since the first determination of the water vapour correction function of the two different Picarro G2401 NIR-CRDS analysers. The coloured lines show the limits for the WMO/GAW compatibility goal (green) and extended (yellow) compatibility goal at an amount fraction of 300 nmol/mol.*

To clarify the fact that different CO standards were used, we added a sentence in section 2.2: "*The CO amount fraction of the standards used for the determination of the water vapour interference ranged from 57 to 741 nmol/mol. No dependency of the water vapour interference on the CO amount fraction was observed.*"

**-page 11, lines 4-9: you state that the clearly decreased bias between TI and PUY instrument was thanks to the drying of sample air for TI. However, you also changed the inlet line and never tested the other line with dryer. Please, mention this uncertainty more clearly in the text.**

Thank you for this comment. It is correct that changing the inlet line causes additional uncertainty. However, there is evidence that the improvement was due to the drying, since parallel measurements were also made for $CH_4$ and $CO_2$ over the same inlet lines, and no change in the bias was observed for these two compounds with and without drying. In addition, we have seen similar results during parallel measurements at other stations of the GAW network. For example, a recent parallel measurement was made at the GAW station Bukit Koto Tabang. In this case, the WCC-Empa TI was equipped with a Nafion dryer for the entire comparison period, but the station analyser was measuring humid air for the first week. The bias was significantly larger during the period when the station analyser was measuring humid air, and decreased after the installation of a Nafion dryer in the same inlet line, as shown in Figure R3.

[Figure]

*Figure R3. Comparison of the Bukit Koto Tabang Picarro G2401 analyser with the WCC-Empa travelling instrument for CO. Time series based on hourly data as well as the difference between the station instrument and the TI is shown. The coloured horizontal areas correspond to the WMO/GAW compatibility (green) and extended compatibility (yellow) goals. The dashed vertical line indicates the time of the installation of the Nafion dryer.*

To clarify that the observed improvement is most likely due to the drying system but also to highlight the potential influence of the inlet system, we added the following sentences on page 11, line 8: "*Potentially, the change of the inlet system could also have been the reason for the reduction in the bias. However, this is unlikely because no change in the bias of $CH_4$ and $CO_2$ amount fraction, which were both measured simultaneously together with CO over the same inlet line, was observed.*"

**-page 11, line 13: did you use the same individual instrument (TI) in PYU and in AMY? Differences between Picarro G2401 individuals, especially regarding CO performance, may be high.**

Yes, the same instrument was used at PUY and AMY (Picarro G2401 CFKADS2098) (see Table 3, instrument serial number is given there).

**-page 11, lines 16-19: since there was three clearly different periods; due the calibrations of AMY instrument; during the comparison measurements at AMY, please provide the biases for each period separately.**

We added the bias for the individual periods. We also realised that the bias of 0.23±8.81 nmol/mol for the entire period given in the discussion paper was wrong, since it referred to a data set that included invalid data. We corrected to 0.10±3.20 nmol/mol.

**-page 12, line 18: is the drying of the sample air the only option, as you state, or is it possible to reach the goal with well and frequently made water vapour tests as well?**

Because sudden changes of the water vapour interference were observed, a correction based on frequent determination of the correction function might still be insufficient. Furthermore, determination of individual correction functions are associated with large uncertainties and drying therefore is recommended.

**You also leave open the questions how to dry and how low water levels are needed to reach. By looking the Figure 10, it looks that water levels below 0.3 %, roughly, would give CO values stable enough when Picarro G2401 instrument is used. That kind of a water levels are possible to reach by using Nafion dryer, for example. The drying issue is an endless story, but open this theme with a few sentences.**

Yes, this is correct. It only is important that the water vapour level remains as constant as possible. Calibration gas and working standards also need to pass through the Nafion dryer to compensate for potential loss in the system.

We added the following sentences at the end of section 3.2:

"*However, this will most likely be insufficient to detect the sudden changes in the correction function that were observed in our experiments.* Consequently, drying of the sample air should be considered when measuring CO with a Picarro G2401 instrument. *Both cryogenic traps and Nafion dryers can be used. WCC-Empa now uses Nafion dryers for the parallel measurements during station audits. Both single tube (MD-070-48S-4) and multi tube (PD-50T-12MPS) Nafion dryers in reflux mode using the Picarro pump for the vacuum in the purge air were successfully used. This reduced the amount of water to approximately 0.06 – 0.22% (single tube) and 0.01-0.03% (multi tube), depending on ambient air humidity. In case of using Nafion dryers, the standard gases must also pass though the dryer to compensate for a potential loss over the dryer.*"

And in the conclusions:

"*This can be implemented through drying of the sample air with cryogenic traps or Nafion dryers.*"

**-figure 7: since the water vapour was probably the main reason for bad agreement between TI and PUY instrument, please add a panel showing the H2O % reading of TI.**

The reason for the bias is the inappropriate compensation of the water vapour interference made by the Picarro instrument. There is a dependency of the bias on the water vapour level, which we showed in Figure 8. Figure R4 shows the original Figure 7 of the paper with an additional panel showing the $H_2O$ reading of the instrument, as requested by the referee. However, we feel that this provides only minor additional information compared to the dependency on the water level shown in Figure 8. We therefore prefer to keep the figure as it is.

[Figure]

*Figure R4. CO comparison at PUY between the WCC-Empa travelling instrument and the PUY Picarro G2401 for the period when the TI sampled humid air. Upper panel: CO time series (1 h data). Middle panel: $H_2O$ measurements of the TI. Lower panel: CO bias of the station analyser vs time. The green and yellow areas correspond to the WMO compatibility and extended compatibility goals.*

**-figure 13: please, mark the calibrations of AMY instrument in lower panel with vertical lines, for example.**

We added vertical lines in the revised version of the paper.

[Figure]

*Figure 1. CO comparison at AMY between the WCC-Empa travelling instrument and the AMY Los Gatos 30-EP QCL analyser. Both instruments sampled dry ambient air. Upper panel: CO time series (1 h data). Lower panel: CO bias of the station analyser vs time. The green and yellow areas correspond to the WMO compatibility and extended compatibility goals. The dashed vertical lines indicate the time of the calibration of the AMY instrument.*

**Technical corrections suggested by Referee #1:**

-page 1, line 20: "analyse" > "analysed".

*Accepted and changed.*

**-page 2, line 1: "GAW programme" is written with and without a capital "p" in this manuscript. Please uniform using lower case "p".**

*Accepted and changed throughout the paper.*

**-page 2, line 11: What is "Empa"? Please add institute full name and location.**

*Done.*

**-page 2, line 11: Use "WCC" instead of "World Calibration Centre". There is often used the full name instead of abbreviation, even though the abbreviation is introduced when first time mentioned. Like three lines below "World Calibration Centre" in written again, and with gas components and instrument techniques is the same issue. Uniform all these issues in the whole manuscript.**

*Done.*

**-page 2, line 14: WCC-$N_2O$, who operates? It is written on page 3, so move it to here.**

*Done.*

**-page 2, lines 28-30: introduction of the GC detectors for CO; now it is not fully clear that CG/FID with methaniser and CG/HgO are two separate instruments to measure CO. Please reword.**

We reworded the sentence to "…, whereas flame ionization detection (GC/FID) in combination with a methaniser and GC with mercuric oxide reduction detector (GC/HgO) *were the two most commonly used techniques for CO measurements* (Zellweger et al., 2009)."

**-page 4, lines 9-10: move the operator of CCL to where it is first time mentioned.**

*Done.*

**-page 5, line 6: add comma after "measurements".**

*Done.*

**-page 6, line 15: maybe "trend in the atmospheric concentrations" over "in the atmosphere".**

Changed to "… due to the significant upward trend *of the $N_2O$ mixing ratio* in the atmosphere…"

**-page 7, lines 13-14: what is the meaning of "(see also Figure, right)"?**

That was a mistake and we deleted it.

References

Brewer, P. J., Kim, J. S., Lee, S., Tarasova, O. A., Viallon, J., Flores, E., Wielgosz, R. I., Shimosaka, T., Assonov, S., Allison, C. E., van der Veen, A. M. H., Hall, B., Crotwell, A. M., Rhoderick, G. C., Hodges, J. T., Mohn, J., Zellweger, C., Moossen, H., Ebert, V., and Griffith, D. W. T.: Advances in reference materials and measurement techniques for greenhouse gas atmospheric observations, Metrologia, 56, 034006, 2019.

NOAA: WMO/IAEA Round Robin Comparison Experiment, https://www.esrl.noaa.gov/gmd/ccgg/wmorr/index.html, last access: 23 July, 2018.

WMO: 19th WMO/IAEA Meeting on Carbon Dioxide, Other Greenhouse Gases and Related Tracers Measurement Techniques (GGMT-2017), Dübendorf, Switzerland, 27-31 August 2017, GAW Report No. 242, World Meteorological Organization, Geneva, Switzerland, 2018.

Zellweger, C., Emmenegger, L., Firdaus, M., Hatakka, J., Heimann, M., Kozlova, E., Spain, T. G., Steinbacher, M., van der Schoot, M. V., and Buchmann, B.: Assessment of recent advances in measurement techniques for atmospheric carbon dioxide and methane observations, Atmos. Meas. Tech., 9, 4737-4757, 2016.

Zellweger, C., Hüglin, C., Klausen, J., Steinbacher, M., Vollmer, M., and Buchmann, B.: Inter-comparison of four different carbon monoxide measurement techniques and evaluation of the long-term carbon monoxide time series of Jungfraujoch, Atmos. Chem. Phys., 9, 3491-3503, 2009.

---

## Author Comment (AC2) · 17 Sep 2019

**Referee #2**

We would like to thank Referee #2 for the valuable comments and her/his time to review the manuscript. The questions/comments/concerns of Referee #2 (bold text) are addressed below. New text in the manuscript is given in italics.

**Specific comments made by Referee #2:**

**1) Throughout the document – the WMO network compatibility goals should be referred to as "network compatibility" rather than just "compatibility" to distinguish the WMO usage from a strict metrological definition of compatibility. Or making the distinction upon first usage if that is preferred.**

We changed to network compatibility goals throughout the entire document.

**2) Page 2, line 11: Define Empa on first usage.**

Done

**3) Page 2, line 14: Define who is the WCC-N$_2$O on first usage.**

Done

**4) Page 4, line 10: "WCC-N$_2$O uses a set of TS traceable to a set of secondary standards …" Are these standards truly secondary (compared directly to the primary standards) or are they actually the normal tertiary standards distributed by the CCL? Or does this "secondary" label relate to the hierarchy internal to the WCC-N$_2$O?**

Thank you for this comment. Yes, it was related to the internal hierarchy. The WCC-N$_2$O hosts a set of tertiary reference standards, which are regularly calibrated by the CCL. Those standards are used to transfer the scale to TS. Therefore, the laboratory standards hosted by WCC-N$_2$O are tertiary. To make that clear the corresponding sentence reads now: "WCC-N$_2$O uses a set of TS traceable to a set of tertiary standards, *which are regularly recalibrated against secondary standards at the CCL.*"

**5) Page 4, line 11-14: Have all of the comparisons been reprocessed onto the current CO_X2014A and N$_2$O_X2006A scales or are they presented on the current scale at the time of the comparison? If the latter is the case I would suggest making the scale explicit by adding a column to tables 1 and 2 showing which scale was actually used. As a follow up I would ask if the comparisons change significantly if reprocessed onto the current scale? This may not be possible within the scope of this paper but would be of interest to data users who would like to use this information to understand potential biases between data from various providers.**

The data has not been reprocessed, and the comparison shows the results using the scales at the time of the audit. We added columns to table 1 and 2 with information on the calibration scales, as suggested by the referee. We added more information on the calibration scales in chapter 2.1, since we were using standards calibrated on the WMO-X2000 CO scale until 2011. However, we only used standards with an amount fraction larger than 185 nmol/mol, and comparisons showed that the two calibration scales agree well at these levels.

*"WCC-Empa continued using the WMO-X2000 calibration scale until 2011 but used only standards with an amount fraction larger than 185 nmol/mol. At these amount fractions, the difference between the WMO-X2000 and WMO-X2004 CO scales are very small and questionably significant within their uncertainties. We therefore consider these two scales as being identical for calibrations made at WCC-Empa."*

We also realised that some of the comparisons involved different calibration scales. However, we decided to keep them since the contribution to the bias due to the scale difference is generally much smaller than the observed variability of all comparisons. The information about the calibration scales used at the stations and the WCCs is available from table 1 and 2. To further highlight a potential bias due to scales differences, we used different symbols in Figures 1 and 4 in the revised paper for the cases with differences in calibration scales.

We also agree that it would be interesting to see if reprocessing onto the current scale would improve the results. However, this is complicated because scale changes may not be linear, and individual reprocessing would have to be applied for each case. This is beyond the scope of the current paper.

**6) Page 5, line 1: When did the parallel measurement approach begin?**

The first parallel measurements were made in 2011. We added this in the revised version of the manuscript.

**6) Page 6, line 28: WMO network compatibility goals are no longer listed as "±".**

Correct. We changed it when we refer directly to the WMO network compatibility goals throughout the manuscript.

**7) Page 7, line 3-6: Have any of the comparisons been reprocessed after the stations have had working standards re-calibrated and drift corrected? It would be very interesting to see if some of these larger offsets are improved with better calibrated standards. This would also provide a more accurate assessment of the bias in the station data but again may be beyond the scope of this paper.**

In some cases, data was reprocessed during or immediately after the audit, but the first assessment was always done without changes to the system. In those cases, only the results of the second comparison are shown, since they better represent the current performance of a station. We added a sentence to clarify this in section 2.1:

"*Since the focus of the paper is on instrument performance, only comparisons involving fully functional instruments were considered. Furthermore, if data has been reprocessed due to any known biases e.g. in working standards, only the results of the final comparison were considered, since they best represent the performance of the measurement instruments at the time of the audit.*"

We further added a footnote in table 1 to clarify that the differences observed during the station audits at Lauder were due to an offset in a working standard and not related to the performance of the FTIR system.

We also agree that a study of the temporal evolution due to reprocessing based on known biases and the impact of multiple audits at stations would be an interesting topic. However, as the referee recognised, this is beyond the scope of the current paper.

**8) Page 8, line 21: Is this statement supported by meta data from the stations, i.e. is there a record of the number of standards used for those early audits that would support this conclusion?**

A SOP for conducting audits is available at https://www.empa.ch/documents/56101/250799/2.pdf/f5a8c0a5-884f-4e0c-b836-96d92dbd260c. The WCCs strictly follow that SOP during audits conducted. Thus, for each audit the meta-information of the number and hierarchy of standards as well the amount fractions of the standards and the scale used for calculating measured ambient air amount fractions is available at the WCCs, also for the early audits.

**9) Page 11, line 7: The figure plots the data as Station – TI, the text has the offset as TI – Station. I suggest changing either the sense of the comparison in this sentence or in the figure to be consistent.**

We changed the sentence to "*The bias of the PUY analyser significantly decreased to 1.20±0.57 nmol/mol (1σ).*" To be consistent, we also changed the sentence on page 10, lines 15-16 to "*During this period, the PUY analyser was measuring on average 5.85±0.94 nmol/mol (1σ) higher than the TI.*"

**10) Page 11, line 12: I think the description of the AMY offaxis_ICOS instrument should be "enhanced performance" off-axis integrated cavity output spectroscopy rather than the stated "cavity enhanced".**

We changed it on page 3, lines 3-4 where it was also wrong, and used the abbreviation OA-ICOS on page 11.

**11) Page 11, line 17-19: I suggest putting in the values for each third of the time period to show how different they are and how much of the variability is due to the calibrations.**

Done

**12) Page 17, Table 1: I suggest listing the CO scale for each comparison if they are not all the same.**

We included this information in the revised manuscript.

**13) Page 19, Table 2: I suggest listing the $N_2O$ scale for each comparison if they are not all the same.**

We included this information in the revised manuscript.

**14) Page 22, Figure 3: As mentioned in the text there are only 2 comparisons of the same FTIR instrument. It might be good to show this by listing n values for each category or at least for the FTIR. I might also suggest keeping the same categories at shown in figure 2 (combining NIR-CRDS and QCL) to be consistent between the two figures but leave this to the author's discretion.**

The number of comparisons is now listed in Figure 3. However, we prefer to keep the results for each single techniques in this figure, since FTIR for example does not appear in any category in Figure 2.

[Figure]

*Figure 3. Boxplot of the slopes uncertainties of the of the regression analysis for the CO performance audits for different analytical techniques including the number of comparisons (n). The horizontal blue line denotes to the median, and the blue boxes show the inter-quartile range.*

**15) Page 24, Figure 7 caption (and other time series plots): The caption says "(1 h data)". I take this to mean the data from both instruments was averaged to hourly averages. If true I suggest making this point clearer.**

Yes, this is correct. We changed the figure caption to "*Comparison of hourly averages of CO at PUY between the WCC-Empa travelling instrument and the PUY Picarro G2401 for the period when the TI sampled humid air. ...*"

**Minor technical corrections suggested by Referee #2:**

All technical corrections suggested by Referee #2 were accepted and changed in the revised manuscript.